# A machine learning-guided adaptive algorithm to reduce the computational cost of integrating kinetics in global atmospheric chemistry models: application to GEOS-Chem versions 12.0.0 and 12.9.1

Lu Shen[1,2], Daniel J. Jacob[2], Mauricio Santillana[3,4], Kelvin Bates[2], Jiawei Zhuang[2], Wei Chen[5]

[1]Department of Atmospheric and Oceanic Sciences, School of Physics, Peking University, Beijing, China
[2]John A. Paulson School of Engineering and Applied Sciences, Harvard University, Cambridge, MA, USA
[3]Computational Health Informatics Program, Boston Children's Hospital, Boston, MA, USA
[4]Department of Pediatrics, Harvard Medical School, Boston, MA, USA
[5]Center for Functional Nanomaterials, Brookhaven National Laboratory, Upton, NY 11973, USA

*Correspondence to*: Lu Shen (lshen@fas.harvard.edu)

**Abstract.** Global modelling of atmospheric chemistry is a grand computational challenge because of the cost of integrating the kinetic equations for chemical mechanisms with typically over 100 coupled species. Here we present an adaptive algorithm to ease this computational bottleneck with no significant loss in accuracy and apply it to the GEOS-Chem global 3-D model for tropospheric and stratospheric chemistry (228 species, 724 reactions). Our approach is inspired by unsupervised machine
learning clustering techniques and traditional asymptotic analysis ideas. We locally define species in the mechanism as fast or slow on the basis of their total production and loss rates, and we solve the coupled kinetic system only for the fast species assembled in a submechanism of the full mechanism. To avoid computational overhead, we first partition the species from the full mechanism into 13 blocks, using a machine learning approach that analyzes the chemical linkages between species and their correlated presence as fast or slow in the global model domain. Building on these blocks, we then pre-select 20
submechanisms, as defined by unique assemblages of the species blocks, and then pick locally and on the fly which submechanism to use in the model based on local chemical conditions. In each submechanism, we isolate slow species and slow reactions from the coupled system of fast species to be solved. Because many species in the full mechanism are important only in source regions, we find that we can reduce the effective size of the mechanism by 70% globally without sacrificing complexity where/when it is needed. The computational cost of the chemical integration decreases by 50% with relative biases
smaller than 2% for important species over 8-year simulations. Changes to the full mechanism including addition of new species can be accommodated by adding these species to the relevant blocks without having to reconstruct the suite of submechanisms.

## 1 Introduction

Global atmospheric chemistry models are computationally expensive because of the need to integrate the coupled kinetic equations describing the model chemical mechanism (Eastham et al., 2018). These mechanisms typically include over one hundred chemical species with lifetimes ranging over many orders of magnitude, requiring the use of high-order implicit solvers to integrate the chemical evolution over model time steps (Brasseur and Jacob, 2017). However, most regions of the atmosphere do not in fact require solving for the full chemical complexity of the mechanism. Here we present an adaptive, stable, and chemically logical (i.e., retaining connections between species involved in the same or similar reactions) algorithm that reduces the computational cost of the chemical integration by half, with losses in accuracy less than 2% and no error growth in multi-year simulations. Our algorithm is based on general chemical principles that can be easily applied to a wide range of mechanisms.

Previous approaches of simplifying atmospheric chemistry mechanisms are reviewed by Brasseur and Jacob (2017). Reducing the dimension of the coupled system can be obtained by decreasing the number of species (Sportisse and Djouad, 2000), isolating long-lived species (Young and Boris, 1977), and removing unimportant reactions (Brown-Steiner et al., 2018). However, the importance of a species or a reaction varies in different atmospheric conditions, so these schemes are not well adapted to global models. Some studies (Jacobson 1995; Rastigeyev et al., 2007) use different subsets of the full chemical mechanism for different regions with specified or locally determined boundaries, but this has limited success because the atmosphere has a continuum of chemical regimes, and geographic boundaries between regimes should be dynamic rather than pre-defined. An adaptive method to define mechanism subsets locally and on the fly has been proposed by Santillana et al. (2010) but the computational overhead of customizing the mechanism on the fly offsets computational gains. The overhead can be avoided by compiling a library of pre-defined mechanism subsets (Shen et al., 2020), but a challenge is to select these subsets in a manner that is chemically logical and portable across mechanisms.

In this work, we continue developing the adaptive method described by Shen et al. (2020). This method pre-assembles a small number of subsets of the full chemical mechanism representing the range of conditions in the troposphere and stratosphere, and selects the most appropriate submechanism to use in the model locally and on the fly. The submechanisms are constructed by first splitting the full mechanism's atmospheric species into $N$ different blocks based on similarity of chemical behaviors, using a machine learning clustering method. We then define the submechanisms as different assemblages of blocks, select $M$ of these assemblages to encompass the majority of chemical conditions in the atmosphere, and build them into the model. The choice of submechanism in the model is then made locally by computing chemical production and loss rates of the mechanism species and deciding which need to be part of the coupled chemical computation ('fast' species) and which can be tracked independently ('slow' species). A major development here is to enforce that chemically connected species be grouped in the same blocks, so that the blocks can be logically modified and extended as the mechanism changes. We further

improve the performance of the method by reducing the number of reactions as well as the number of species in the submechanisms.

## 2 Method description

Here we describe the adaptive method as applied in the GEOS-Chem global model, although it is applicable to any model. We begin with a brief description of the model as relevant to the presentation.

### 2.1 GEOS-Chem model

We use the GEOS-Chem version 12.0.0 global 3-D model for tropospheric and stratospheric chemistry (https://doi.org/10.5281/zenodo.1343547) with 12 CPUs in a shared-memory Open Message Passing (Open-MP) parallel
environment. For development and testing purposes, we choose a horizontal resolution of 4°×5° and 72 pressure levels extending from surface to 0.01 hPa and drive the model with NASA MERRA2 assimilated meteorological data. The full mechanism for oxidant-aerosol chemistry in the model has 228 species and 724 reactions, including coupled gas-phase and aerosol chemistry for the troposphere and stratosphere (Sherwen et al., 2016; Eastham et al., 2014). The chemical operator uses a 4th-order Rosenbrock implicit method, implemented through the Kinetic Pre-Processor (KPP) (Sander and Sandu, 1996),
to solve for the chemical evolution of species concentrations, involving iterative calculations and inversion of the Jacobian matrix that stores the sensitivity of species tendencies (production minus loss rates) to concentrations. In the simulations presented here, methane, $N_2O$, and other long-lived halocarbons have fixed concentrations in surface air (Eastham et al., 2014; Murray, 2016) so that the longest resolved chemical modes are less than a year.

As part of this study, we test the portability of our adaptive algorithm by moving it from GEOS-Chem version 12.0.0 to GEOS-
Chem version 12.9.1 (https://doi.org/10.5281/zenodo.3950473). This new version of GEOS-Chem has a thoroughly updated mechanism of 262 species and 850 reactions, including improved organic nitrate chemistry (Fisher et al., 2018), isoprene chemistry (Bates and Jacob, 2019), and halogen chemistry (Wang et al., 2019). From version 12.0.0 to 12.9.1, we need to remove 49 old species and add 83 new species.

### 2.2 Separation of fast and slow species and reactions

Coupling between species in the Rosenbrock chemical solver is needed only for species with sufficiently fast production or loss rates (fast species), and similarly reactions need to be considered only if they are sufficiently fast. We separate the atmospheric species as fast or slow based on their production and loss rates relative to a threshold $\delta$: fast if either $P_i(\boldsymbol{n}) \geq \delta$ or $L_i(\boldsymbol{n}) \geq \delta$, slow if $P_i(\boldsymbol{n}) < \delta$ and $L_i(\boldsymbol{n}) < \delta$ ($P_i$ and $L_i$ refer to the production and loss rates of the $i^{th}$ species, $\delta$ is a threshold, and $\boldsymbol{n}$ is a vector of concentrations of all species). To get a sense of a relevant threshold, consider the hydroxyl radical (OH) which
is central to driving oxidant-aerosol chemistry. OH has a daytime concentration of the order of $10^6$ molecules cm$^{-3}$ and a

lifetime of 1s, so its production and loss rates are of the order of $10^6$ molecules cm$^{-3}$ s$^{-1}$. Species with production and loss rates smaller than $10^2$-$10^3$ molecules cm$^{-3}$ s$^{-1}$ are unlikely to have fast influence on other species in the mechanism (Santillana et al., 2010; Shen et al., 2020). In this study, we use $\delta$ from 500 to 1500 molecules cm$^{-3}$ s$^{-1}$ to partition the fast and slow species. We also define species with chemical lifetime longer than 10 days as long-lived.

We pre-select a limited number ($M$) of submechanisms for which we pre-code the Jacobian matrix needed by the Rosenbrock solver in KPP. In each submechanism, if a reaction is slower than 10 molecules cm$^{-3}$ s$^{-1}$ for all gridboxes that select this submechanism, then the reaction is considered negligible and removed from the submechanism. The logic is that such a slow reaction will not contribute significantly to the total species production/loss rate threshold $\delta$ =500-1500 molecules cm$^{-3}$ s$^{-1}$. About 40-60% reactions can be removed using this strategy without incurring significant error. For example, reactions of short-
lived volatile organic compounds (VOCs) are removed in stratospheric gridboxes, and daytime photochemical reactions are removed in nightime gridboxes. Tests indicate that increasing the reaction rate threshold to 100 molecules cm$^{-3}$ s$^{-1}$ incurs significant error.

We solve for the fast species in their submechanism using the standard Rosenbrock solver. For the slow or long-lived species, we use instead an explicit analytical solution that assumes first-order loss (Santillana et al., 2010), written as

$$\frac{dn_i}{dt} = P_i - L_i = P_i - k_i n_i \tag{1}$$

$$n_i(t + \Delta t) = \frac{P_i(t)}{k_i(t)} + \left(n_i(t) - \frac{P_i(t)}{k_i(t)}\right)e^{-k_i(t)\Delta t} \tag{2}$$

where $n_i$ is the concentration of species $i$, $P_i$ and $L_i$ are the production and loss rates, $k_i$ is the rate coefficient of the first-order loss, and $\Delta t$ is the time step. Solving for Eq.2 entails negligible computational cost.

**2.3 Defining the distance between species in the mechanism**

We construct subsets ('blocks') of the species in the mechanism species based on their linkages through the mechanism reactions. This is done by defining the species distances in the mechanism using graph theory. In general, two species should have shorter distances if they appear together in multiple reactions (e.g. NO and NO$_2$, HO and HO$_2$) or have similar products in the mechanism. From the full mechanism of 228 species and 724 reactions, we find 3400 species pairs of reactants-products and map them to an undirected graph that has 228 vertices and 1422 edges. For example, in the reaction A+B-→C, there are
2 pairs (A-C and B-C) of reactants-products, 3 vertices (A, B, and C) and 2 edges (A-C and B-C). If species $i$ and $j$ share the same edge, we define their distance as

$$D_{i,j} = \frac{T_{i,j}}{\sqrt{T_i T_j}} \tag{3}$$

Where $T_{i,j}$ is the number of reactions that include both species $i$ and $j$ (with $i$ as reactant and j as product, or $i$ as product and j as reactant), and $T_i$ (or $T_j$) is the number of species that appear in the same reactions with species $i$ (or $j$). If species $i$ and $j$ never appear in the same reaction so they do not share the same edge in the graph, their distance is calculated as the length of the shortest path from species $i$ to $j$. For example, the distance of toluene (TOLU) and xylene (XYLE) can be defined as the length of path TOLU-GLYX-XYLE (Figure 1, GLYX is glyoxal). Similarly, we can also define the distance between two blocks using Eq.3, in which we define $T_{i,j}$ as the number of reactions that include species in block $i$ and $j$ (one is the reactant and the other is the product) and $T_i$ (or $T_j$) as the number of blocks that have reactions with block $i$ (or $j$).

This definition of distance between species does not take into account of the rates of individual reactions connecting two species, and thus may overestimate weak links resulting from slow reactions. Accounting for relative reaction rates in a general definition of distances would introduce complication because the rates depend on the local chemical environment. We tried weighting species distances by the logarithms of global mean reactions rates but found no significant effects on results.

Equation 3 can define the distance of species along reaction chains, but it may overestimate the distance of species that do not react with each other but have similar products (e.g. XYLE and TOLU). These species usually come from the same chemical family and should be close to each other in terms of distances. In our work, we address this shortcoming as follows. First, we denote each species $i$ by a vector ($D_i$) that contains its distance with all other species. The similarity of two species $i$ and $j$ can be thus defined as their Euclidean distance $\|D_i - D_j\|$. Second, for each species $i$, we decrease its distance with the 5 species that have highest similarity with it by 50% and this scaling is applied only once for each species pair. The logic is that the number of species with similar chemical characteristics is usually around 5 and decreasing the distances among them by 50% can increase the probability of these species to be in the same chemical blocks after the optimization process. We carried out a number of tests by perturbing the parameters used here and examine if the optimized chemical blocks are chemically logical, and results show that using 10 highest-similarity species instead of 5 or decreasing distances by 30%-70% instead of 50% did not significantly change the results. We store these modified distances of all species pairs in a 228x228 matrix.

**2.4 Selection of species blocks and submechanisms**

We construct submechanisms by assemblage of blocks in order to minimize the number of fast species to be integrated with the Rosenbrock solver in the model. To partition the species into $N$ blocks, we use a training dataset from a GEOS-Chem simulation for 2013 consisting of the first 10 days of February, May, August, and November sampled every 6 hours (160 time steps in total).

For the 228-species mechanism in GEOS-Chem, there are in $2^{228}-1$ possible combinations of species and we need to pre-select $M$ of them to form submechanisms that can encompass the range of atmospheric conditions. To reduce the dimensionality of

this problem, we start by splitting the 228 species into $N$ different blocks. A block is considered as fast if at least one species in that block is fast ($P$ or $L > \delta$). Building on the $N$ blocks, we define each submechanism as an assemblage of fast blocks, which yields $2^N - 1$ possible submechanisms. Each gridbox in the model domain may correspond to one of these $2^N - 1$ submechanisms. More specifically, for each gridbox $j$, we diagnose species $i$ as fast or slow following the definition of Section 2.2. We define $y_{i,j} = 1$ if any species in the block is fast or $y_{i,j} = 0$ if all species in the block are slow. Thus, the fraction $Z_1$ of all species that needs to treated as fast can be written as

$$Z_1 = \frac{1}{\Omega} \sum_j \sum_i y_{i,j} \tag{4}$$

where $\Omega$ is the number of species × gridboxes.

We need to limit the number of submechanisms to a small number $M$ in order to keep the compilation of the code manageable. Gridboxes that do not correspond to any of the $M$ submechanisms need to be matched to one of the $M$ submechanisms by moving some blocks from slow to fast, and we select the submechanism that has a minimum number of moves. As such, the values of some $y_{i,j}$ need to be changed from 0 to 1 and we refer to $y^*_{i,j}$ as the indicators adjusted by these changes. The fraction of species $f(M, N)$ that need to be treated as fast over the global domain is given by:

$$f(M,N) = \frac{1}{\Omega} \left( \sum_{V_1} \sum_i y_{i,j} + \sum_{V_2} \sum_i y^*_{i,j} \right) \tag{5}$$

where $V_1$ are the gridboxes that can be represented directly by the $M$ chemical submechanisms, and $V_2$ are the gridboxes that must be matched to the $M$ submechanisms.

The cost function $Z$ to be minimized in the selection of submechanisms can be written as

$$Z = f(M,N) + \gamma Dist \tag{6}$$

Where $Dist$ is the sum of distances for all pairs of species if they are in the same block, $\gamma$ is a regularization factor; $f$ is the fraction of species that needs to be treated as fast over the testing domain based on $M$ and $N$ (Eq. 5). We adjust $\gamma$ so that the second term on the right part of Eq.6 contributes to 20% of the total cost function. We seek the partitioning of species into blocks that will minimize $Z$, and we use for that purpose the simulated annealing algorithm (Kirkpatrick et al., 1983). We tested a range of values from 5 to 20 for $N$ and from 10 to 40 for $M$. In the simulated annealing algorithm, we start from a randomly generated partition of the $N$ blocks. In each iteration, we randomly move one species from one block to another. If the cost function decreases, this transition is accepted; otherwise, it is accepted with a probability controlled by a parameter named temperature. The temperature parameter decreases gradually as the optimization proceeds (Kirkpatrick, 1983).

The explicit solution by Eq. 3 does not strictly conserve mass (Shen et al., 2020), and Shen et al. (2020) previously found that
this is a problem for halogen species in the stratosphere due to the long lifetime of the collective halogen families and the alternance of the component species as fast and slow over day and night. To avoid this problem, we treat all 37 reactive inorganic halogen species as fast in the stratosphere. Thus, among the $N$ blocks, 2 are allocated to the reactive inorganic halogen species, and $N$-2 are allocated to the other species. The transitions of species between the 2 inorganic halogens blocks and the other $N$-2 blocks are not accepted in the optimization process.

**2.5 Error analysis**

We use the Relative Root Mean Square Error (RRMSE) metric as given by Sandu et al. (1997) to characterize the error in our reduced mechanism:

$$RRMSE_i = \sqrt{\frac{1}{Q_i} \sum_{j=1}^{Q_i} \left( \frac{n_{i,j}^{\text{reduced}} - n_{i,j}^{\text{full}}}{n_{i,j}^{\text{full}}} \right)^2} \qquad (7)$$

where $n_{i,j}^{\text{reduced}}$ and $n_{i,j}^{\text{full}}$ are the concentrations for species $i$ and gridbox $j$ in the reduced and full chemical mechanisms, and the
sum is over the $Q_i$ ordered gridboxes that account for 99% of the global mass of species $i$, We calculate separate RRMSEs for the boundary layer (surface to 2 km), free troposphere (2 km to tropopause), and stratosphere with the same 99% threshold in each atmospheric domain. As a test, we also calculate the RRMSEs over the gridboxes that can account for 99% of the mass in each atmospheric domain (the 99% thresholds are different in different domains in this case).

A second metric to evaluate our adaptive chemical mechanism is the relative difference of global atmospheric masses for
individual species compared to the standard simulation. This tests for accumulating bias over long simulation periods.

**3 The adaptive algorithm for the chemical operator**

**3.1 Potential for local simplifications of atmospheric chemistry mechanisms**

Figure 2 displays the potential for local simplification of the full mechanism over the global domain, based on local chemical production and loss rates for the 228 species simulated by GEOS-Chem. Using a threshold $\delta$ of 500 molecules $cm^{-3}s^{-1}$ for
production and loss rates to define the fast and slow species (see Section 2.2 for the selection of this threshold), a given percentage of species can be excluded from the coupled chemical mechanism. That percentage is 75% for surface grid cells and reaches 90% in the stratosphere. When compared with removing long-lived species (lifetime > 10 days), a strategy that is most commonly used in simplifying the chemical mechanism (e.g., Young and Boris, 1977), removing slow ones is more effective because it can exclude a large majority of unimportant species. As seen from Figure 2a, long-lived but fast species

are only present in the lower troposphere and their percentage is below 1% when averaged globally. Figure 2b shows the percentage of slow reactions ($<10$ molecules $cm^{-3}s^{-1}$) in the atmosphere, which is found to be 75-85% in the troposphere and 90% in the stratosphere (Figure 2b). A slow reaction does not necessarily mean that it is not important, but if it is slow in all gridboxes of a subdomain of the atmosphere then we can safely remove it in this subdomain. These results show that most of the atmosphere does not in fact require solving for the full complexity of the mechanism, so considerable simplification is possible if we can recognize the spatial and temporal patterns of chemical complexity in different atmospheric subdomains. As we will show later, we are able to exclude 50-80% species and 40-60% reactions at different altitudes of the atmosphere from the coupled system in our adaptive algorithm (Figure 2).

## 3.2 Performance of our adaptive algorithm

Our work addresses two problems in the original Shen et al. (2020) approach. First, the blocks identified by their machine learning approach based solely on minimizing computational time (Equation 6 with no regularization term) were not chemically logical. Some species known to be chemically coupled by simple inspection of the mechanism were separated in different blocks. The regularization term addresses this shortcoming by penalizing the separation of species that are linked in the mechanism by direct and indirect reactant-product relationships. Second, Shen et al. (2020) only achieved 30-40% time-savings. Here we improve the performance of the algorithm by not only isolating slow species but also removing slow reactions from the submechanisms, thus speeding up the computation of the Jacobian. The slow reactions removed in each submechanism are pre-defined (see Section 2.2 for more details).

Figure 3 shows the fraction of fast species that needs to be solved using the chemical solver in the global domain as a function of $M$ (submechanisms) and $N$ (blocks). If $N$ is low so each block is large, the mixing of slow species with fast ones will increase the likelihood of treating all species in this block as fast. If $N$ is too high relative to $M$, more gridboxes cannot be represented by the $M$ submechanisms and hence have to use submechanisms of higher complexity than needed. For each $N$, there exists a threshold for $M$ above which the cost function remains almost unchanged. In order to make the code manageable, we choose to use $M = 20$ resulting in an optimal value $N = 13$ at which only 30% of the species need to be treated as fast in the global tropospheric and stratospheric domain (Figure 3). As shown in Figure 3, this performance is relatively insensitive to the choice of $M$.

Figure 4a-b shows the method and the results of partitioning of species into the 13 ($N$=13) blocks (the detailed list of species is in Table 1). Oxidants and methane oxidation products are important everywhere so blocks 1 and 2 are part of the submechanism in 50-80% of gridboxes (Figure 4b). Aside from the oxidants, bromine and chlorine radicals (block 3) also play a pervasive role in tropospheric and stratospheric chemistry, and are part of the submechanism in 39% of gridboxes (Figure 4b). Our algorithm can also largely separate anthropogenic VOCs from biogenic ones, although a few such species may overlap because they have similar products (e.g. block 7 contains both anthropogenic and biogenic precursors of glyoxal; see Table 1).

Anthropogenic VOC species are important in 10-20% gridboxes, which are mainly found in the lower troposphere (Figure S1). Biogenic VOC species generally have shorter lifetimes, so they are found to be important only in 0.5-4% gridboxes in the terrestrial lower troposphere near their sources (Figure S2). Most of the secondary organic aerosols can be found in Block 8 and 11, which are found to be fast in 0.5-3% gridboxes (Figure 4b). Halocarbons are relatively inert in the atmosphere and they are found to be important in <2 % of gridboxes (Figure 4b).

Figure 4c shows the network of these 13 blocks in the full mechanism. A connection between two blocks means that species from these two blocks are reactants or products in the same reactions. If more species from two blocks are found in the same reactions and have similar products, the distance between these two blocks is shorter (Eq.3), as represented by the length of edges in the graph. As seen from the figure, atmospheric oxidants play a central role in the mechanism; thus they connect with all other blocks. Anthropogenic and biogenic VOCs have similar products (e.g. acetone and formaldehyde) and they are found to be interconnected with each other. Halogen species interact with the system mainly through the atmospheric oxidants. This network also shows that the optimized blocks by our algorithm are chemically logical.

Figure 5 shows the composition of the 20 submechanisms as defined by the 13 blocks. The first 11 submechanisms do not need to solve any biogenic VOC species and include <40% of the full mechanism's species (Figure 5a). More than 70% of gridboxes select these non-biogenic submechanisms, which are mainly distributed in the stratosphere and free troposphere (Figure 5b and S3). The other 9 submechanisms have higher complexity and are mainly used in the lower troposphere over the continents (Figure 5b and S3). Only 0.05% of gridboxes need to use the full chemical mechanism.

Based on different choices of the rate thresholds $\delta$ separating fast and slow species, we can adjust the complexity and accuracy in the adaptive mechanism. Increasing the threshold can speed up the computation but at the expense of accuracy. Figure 6 shows the median RRMSE (see the definition in Eq.7) of all species and the CPU time used by chemical integration for threshold rates of 500 and 1500 molecules cm$^{-3}$ s$^{-1}$, compared to the full chemical mechanism. This comparison is conducted by running the simulation for 3 years to examine the sensitivity to different $\delta$. 3 years exceeds the longest chemical modes in our simulation (Section 2.1). For each $\delta$, we test the effects of using two strategies, including isolating slow species (A1) and removing slow reactions (A2) (see Figure 6). By isolating slow species (A1), we can reduce the chemical integration time by 38-43% with median errors of 0.6-1.3% among all species of all gridboxes. By further removing the slow reactions in each submechanism (A1+A2), we can reduce the CPU time by 44-49% and the median RRMSEs for the full atmospheric domain remain at 0.9-1.5%. The median RRMSEs are <0.4% in the boundary layer and 0.8-2.0% in the free troposphere and stratosphere. When using a higher threshold $\delta = 1500$ molecules cm$^{-3}$ s$^{-1}$ to isolate slow species and removing the slow reactions, we can reduce the chemical integration time by 50%, and the median RRMSE is smaller than 2% for the full atmospheric domain (smaller than 0.5% in the boundary layer). The relative error on concentrations compared to the standard simulation is below 0.5% in tropical and mid-latitude regions for key species like O$_3$, OH, sulfate and NO$_2$, and could be higher (1-6%) in

high latitudes where more chemical complexity reduction happens (Figure S4). Using a higher threshold of $\delta$ (> 1500) only leads to marginal improvement in computer time but the RRMSE quickly increases.

Figure 7 shows the evolution of the RRMSE over an 8-year period for all 228 individual species in the mechanism, using a $\delta$ of 1500 molecules $cm^{-3}$ $s^{-1}$ to isolate slow species and also removing the slow reactions. The results are shown in different atmospheric domains including the boundary layer, free troposphere, and stratosphere. There is no significant growth in error over the 8-year period. The RRMSEs for key species including ozone, OH, sulfate and $NO_2$ are smaller than 0.5% and are within ±10% for >95% of the other species in the boundary layer (Figure 7a). The median RRMSEs are higher (1-2%) in the free troposphere and stratosphere where most of the reduction of chemical complexity occurs (Figure 7b-c). The median RRMSE in the stratosphere increases slightly in the first 20-30 months and then stabilizes, reflecting the long time scale for chemical aging to abate the sensitivity to initial conditions. We also calculate the RRMSEs for each species in the three atmospheric domains using three different thresholds so that we can account for 99% mass in each domain (Figure S5). This means more gridboxes with lower species concentrations will be accounted for in the calculation so the RRMSE are slightly higher (0.4% in the boudanry layer and 2-3% in the free troposphere and stratosphere). Table S1 lists the species with 10% highest RRMSE in each of the three atmospheric domains, dominated by secondary VOCs and iodine radicals in the troposphere, and VOC species in the stratosphere. None of these species play a central role in the chemistry for the corresponding atmospheric domains.

Figure 8 shows the relative differences of global atmospheric masses over the 8-year simulation in the boundary layer, free troposphere, and stratosphere. The relative differences are within 10% for >99% of the species in troposphere and for >95% of the species in the stratospehre. Species with the largest errors are inorganic halogens and VOC species (more details can be found in the boxplots in Figure S6 and S7). Table S1 lists the species with 10% highest relative bias in atmospheric masses; all have minor importance in  atmospheric chemistry. OH has a bias <0.2% in the troposphere and <0.01% in the stratosphere. Other key species like ozone and sulfate have a relative difference <0.5% in the troposphere and <0.1% in the stratosphere (Figure S8). The relative difference for $NO_2$ in the stratosphere changes slightly from 0% to -0.6% in the first 30 months and then stabilizes at -0.6% (Figure S8).

### 3.3 Adapting to mechanism updates

Chemical mechanisms in models are frequently updated, including addition and removal of species. Our algorithm can accommodate mechanism updates without requiring reconstruction of the submechanisms. New species simply need to be added to the appropriate blocks. Figure S9 shows the diagram for adding new species into the mechanism. Attribution of a species to a given block can be easily determined by its chemical behavior and the percentage of gridboxes that treat this species as fast when averaged globally. In order not to compromise the computational efficiency, the basic rule is to not mix faster species with slower ones. For example, biogenic VOC species and their products could go to Block 8-9 if the percentage

of gridboxes that treat them as fast is >1% or Block 10-11 if the percentage is <1%. Our algorithm is robust to misplacements of new species, which may affect computational performance but will not enlarge the error.

To demonstrate this procedure, we ported our method originally developed with the GEOS-Chem 12.0.0 chemical mechanism (228 species and 724 reactions) to the more recent GEOS-Chem 12.9.1 version (262 species and 850 reactions). This involved major changes to the mechanism including for organic nitrate chemistry (Fisher et al., 2018), isoprene chemistry (Bates and Jacob, 2019), and halogen chemistry (Wang et al., 2019), with removal of 49 species and addition of 83 new ones. We add these new species following the diagram in Figure S9. After running the new version of the model for 12 months, our reduced

algorithm shows consistent improvement in performance, reducing the chemical integration time by 53% and maintaining error of 1.2% in the atmosphere and <0.5% in the boundary layer (Figure 6c).

## 4. Conclusions

The high computational cost of chemical integration is a long-standing limitation in global atmospheric chemistry models. Typical chemical mechanisms include over 100 species coupled on short time scales. Previous research has proposed a variety

of ways to speed up the chemical operator, all involving some loss of accuracy or generality. In this study, we have presented a machine learning-guided adaptive method that can reduce the chemical integration time by 50% when compared to the full chemical mechanism while maintaining error at the level of 2% and retaining full diagnostic capability.

In our algorithm, we first partition the mechanism species in into chemically logical blocks using a machine learning approach that analyzes production/loss rates and chemical linkages between species. We then assemble these blocks into an ensemble

of submechanisms to encompass the range of chemical environments in the atmosphere. The model picks locally on the fly which submechanism to use based on species' production and loss rates. The original mechanism can thus be greatly reduced in most environments while maintaining complexity where needed. Our method can reduce the chemical integration time by 50% while incurring errors of less than 2%, with no error growth over multi-year global simulations. Updates to the original mechanism can be accommodated by assigning new species to the existing blocks without having to reconstruct the suite of

submechanisms.

Our method has many advantages over previously proposed approaches to reduce chemical mechanism: (1) it is chemically logical; (2) it can save 50% computer time in chemical integration with errors lower than 2%; (3) it is stable (no error growth over time) for 8-year simulations; (4) it retains full diagnostic information of concentration and rates; and (5) it is scale-independent. Our algorithm can significantly ease the computational bottleneck of chemical kinetics in global atmospheric

chemistry models.

**Code availability**. The standard GEOS-Chem code is available through https://doi.org/10.5281/zenodo.1343547 (version 12.0.0) and https://doi.org/10.5281/zenodo.3950473 (version 12.9.1). The updates for the adaptive mechanism can be found at https://doi.org/10.7910/DVN/KASQOC.

**Data availability**. All datasets used in this study are publically accessible at https://doi.org/10.7910/DVN/KASQOC.


**Author contribution.** L. Shen and D. Jacob designed the experiments and L. Shen carried them out. L. Shen and D. Jacob prepared the manuscript with contributions from all co-authors.

**Competing Interests**. The authors declare that they have no conflict of interest.


**Acknowledgments.** This research has been supported by the NASA Modeling and Analysis Program (NASA-80NSSC17K0134) and by the US EPA Science to Achieve Results (STAR) Program (EPA- G2019-STAR-C1).

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

**Figures and Tables**

Table 1. Partitioning of GEOS-Chem chemical species into $N = 13$ blocks[a].

| Categories | Blocks | Major components | Species | %gridbox[b] |
|---|---|---|---|---|
| Oxidants and methane products | 1 | Oxidants | MPN, N2O5, HNO3, O3, NO2, MO2, H2O, NO3 | 74.3±14.5% |
| | 2 | Oxidants, methane | HNO4, HNO2, H, CH4, H2O2, CH2O, HO2, NO, O, CO, O1D, OH | 55.3±11.6% |
| Inorganic halogens | 3 | Bromine and chlorine radicals | BrNO2, IONO, OIO, ClOO, OClO, BrCl, HOI, Br2, IONO2, BrNO3, I, IO, HOBr, HOCl, ClNO3, BrO, HCl, HBr, Cl, Br, ClO | 39.4±18.1 |
| | 4 | Iodine reservoirs | AERI, ISALA, ISALC, I2O4, I2O2, I2O3, IBr, INO, HI, ICl, Cl2O2, ClNO2, BrSALC, BrSALA, I2, Cl2 | 1.7±1.4% |
| Anthropogenic VOCs and sulfate | 5 | Alkanes, alkenes, acetone, sulfur compounds | MSA, MAP, ETP, DMS, PAN, SO4, ATOOH, MP, C2H6, ATO2, ACET, ETO2, ALD2, MCO3, SO2 | 20.0+9.1% |
| | 6 | Higher alkanes and oxidized organics | PPN, RA3P, RB3P, RP, ALK4, R4P, C3H8, EOH, A3O2, B3O2, RCO3, KO2, ACTA, MGLY, R4O2, R4N2, RCHO, MEK | 9.5±4.1% |
| | 7 | Aromatics, glyoxal, and related OVOCs | SOAGX, IMAE, DHDC, BENZ, TOLU, TRO2, BRO2, XRO2, XYLE, HPALD, DHPCARP, HPC52O2, GLYX, HCOOH, GLYC, HAC | 3.9±1.7% |
| Biogenic VOCs | 8 | Isoprene products (low NOx), secondary organic aerosols | LVOCOA, LVOC, SOAIE, SOAME, IEPOXD, IEPOXA, IEPOXB, HC187, IAP, VRP, MOBA, DHMOB, RIPB, RIPA, RIPD, IEPOXOO, HC5OO | 2.5±1.4% |
| | 9 | Isoprene, isoprene nitrates | IMAO3, PP, MRP, DIBOO, IPMN, INPN, ISOPNB, MVKOO, CH2OO, PO2, ISOPNDO2, MACROO, ISOP, LIMO2, ISOPNBO2, ISOPND, VRO2, ISN1, HC5, RIO2, INO2, MRO2, PRPE, MACR, MVK | 3.8±2.0% |
| | 10 | Terpenes | INDIOL, MONITA, IONITA, PIP, HONIT, ISNP, MTPA, MTPO, MOBAOO, LIMO, ROH, MONITS, CH3CHOO, MVKN, MONITU, MGLOO, R4N1, OLND, OLNN, PIO2 | 3.0±1.5% |
| | 11 | Isoprene products (high NOx), secondary organic aerosols | ISN1OA, ISN1OG, PYAC, SOAMG, DHDN, PMNN, PRPN, MAOP, ETHLN, ISNOHOO, NPMN, ISNOOB, MACRNO2, GAOO, MGLYOO, PRN1, PROPNN, MAN2, ISNOOA, MACRN, MAOPO2, NMAO3 | 0.5±0.6% |
| Organic halogens and other long-lived species | 12 | Halocarbons | CH2I2, CH2ICl, CH2IBr, CH3CCl3, CH3I, CHBr3, CH2Cl2, CHCl3, CH2Br2, HCFC123, HCFC141b, HCFC142b, HCFC22, CH3Br, CH3Cl | 0.47±1.70% |
| | 13 | Chlorofluorocarbons | H1301, H2402, CCl4, CFC11, CFC12, CFC113, CFC114, CFC115, H1211, N2O, N, OCS | 0.55±1.91% |

[a]The full GEOS-Chem mechanism has 228 species. The full names of these acronyms can be found at http://wiki.seas.harvard.edu/geos-chem/index.php/Species_in_GEOS-Chem.

[b]Percentage of gridboxes in the global tropospheric+stratospheric domain that treat this species block as fast. We use a threshold $\delta$ of 500 molecules cm$^{-3}$ s$^{-1}$ to partition the fast and slow species.

### Definition of species distance between TOLU and XYLE

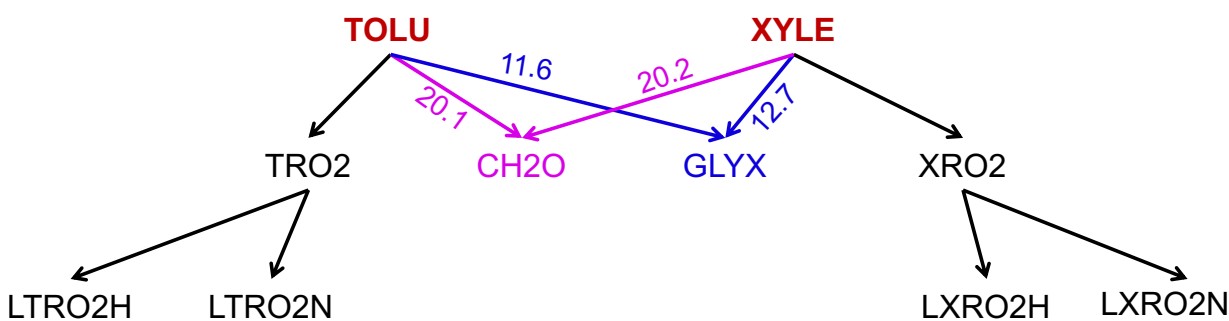

**Figure 1**. Definition of species distances for TOLU (toluene) and XYLE (xylene) using the analysis of family trees in graph
theory. The number denotes the distance between species as calculated by Eq. 3. The shortest path from TOLU to XYLE is
TOLU-GLYX-XYLE in this graph, where GLYX is glyoxal.

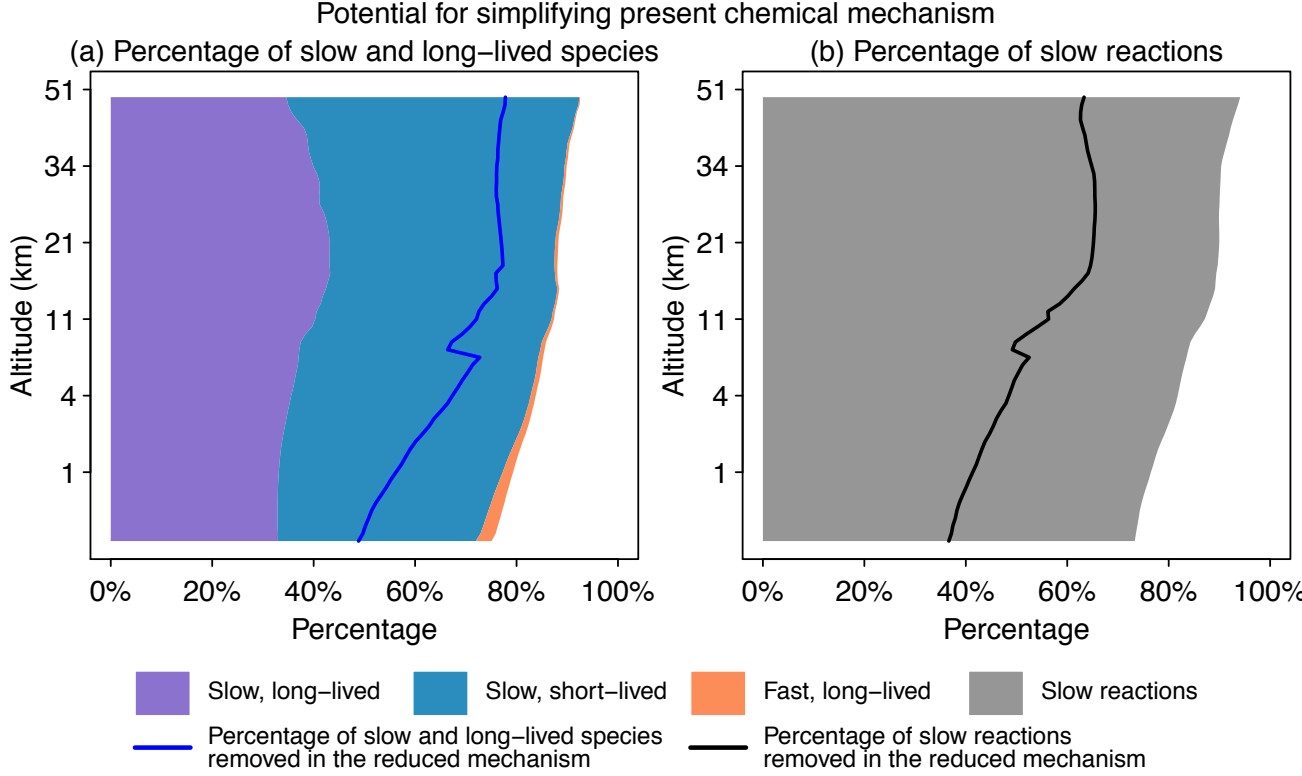


**Figure 2**. **Potential for simplifying the full chemical mechanism in a global GEOS-Chem model simulation.** Panel (a) shows the percentage of slow and long-lived species by altitude when averaged globally on Aug 1$^{st}$ 2013 at 0 GMT. We use a threshold of 500 molecules cm$^{-3}$ s$^{-1}$ to partition fast (*P* or *L* is > 500 molecules cm$^{-3}$s$^{-1}$) and slow species (*P* and *L* are both < 500 molecules cm$^{-3}$s$^{-1}$), and a lifetime of 10 days to separate long-lived and short-lived species. The blue line denotes the

percentage of slow and long-lived species that are actually removed in the reduced mechanism. Panel (b) shows the percentage of slow reactions (<10 molecules cm$^{-3}$s$^{-1}$) by altitude. The black line is the percentage of slow reactions actually removed in the reduced mechanism.

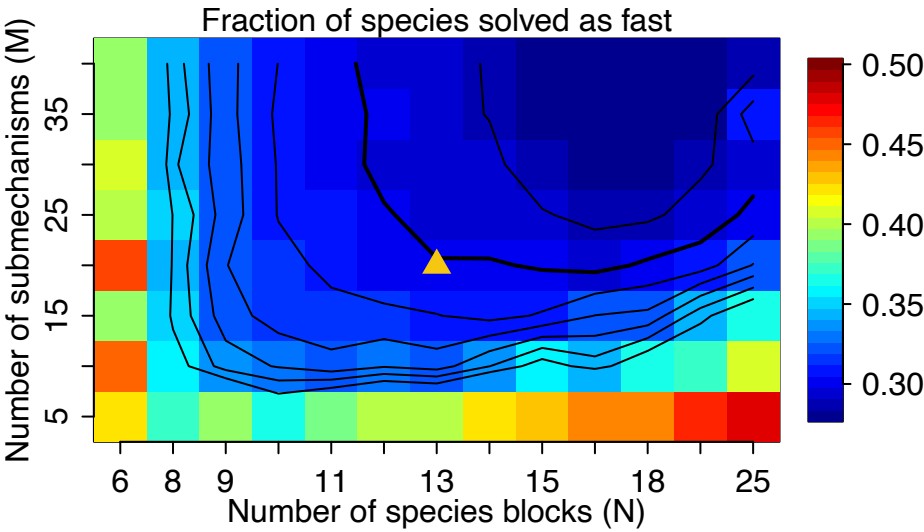

**Figure 3**. The fraction of species solved as fast as a function of *M* and *N*. We use *M*=20 and *N*=13 in our work, as shown by the triangle in the figure, with a threshold δ of 500 molecules cm$^{-3}$ s$^{-1}$ to partition the fast and slow species. The contour lines are spaced by 0.01 with the bold line for 0.30.



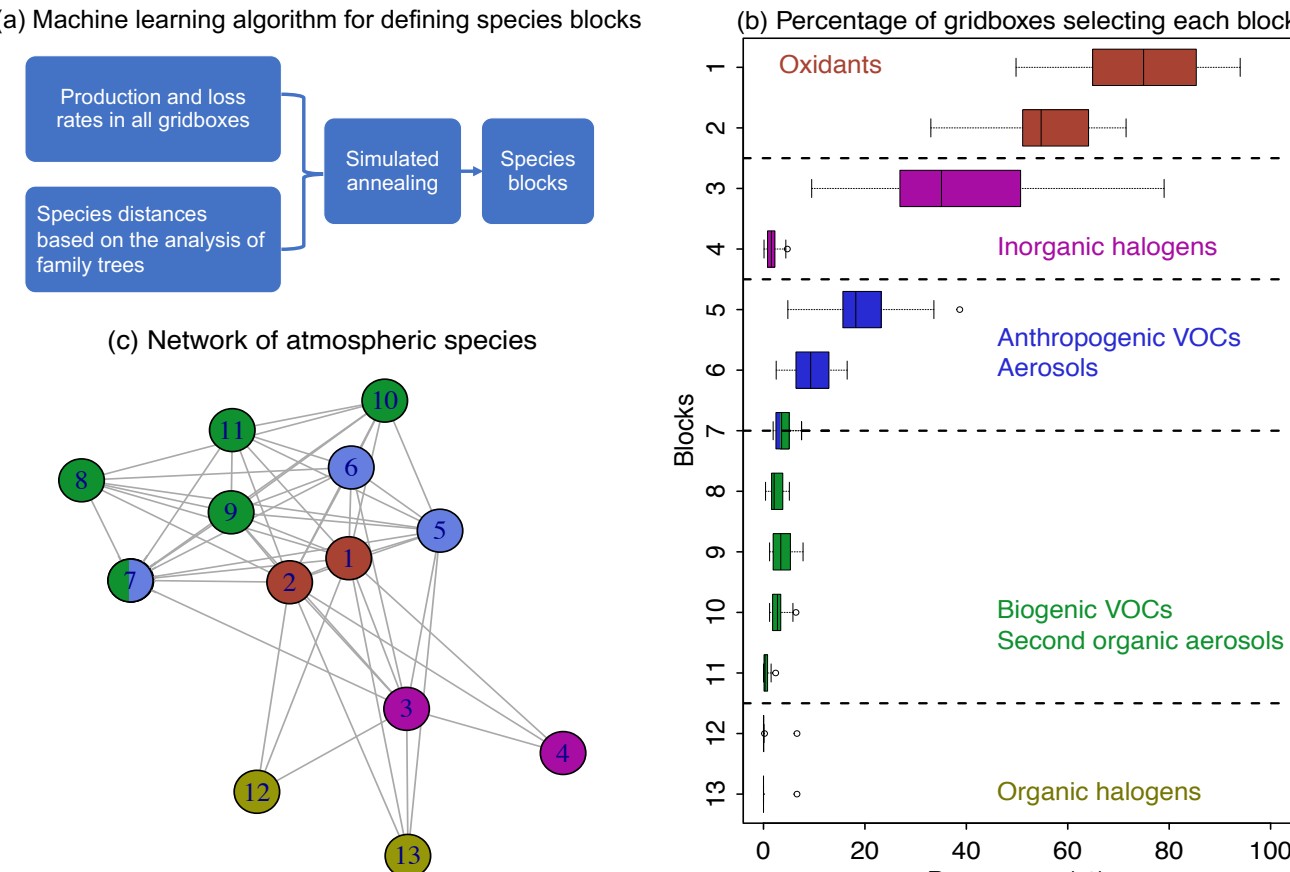

**Figure 4. Optimized species blocks and their network in the full chemical mechanism.** Panel (a) describes the machine learning method to solve for the species blocks. See more details in Section 2. Panel (b) shows the 13 species blocks and the percentage of gridboxes that treat the blocks in their submechanisms. The list of species in each block is given in Table 1. Block 7 includes both anthropogenic and biogenic VOCs. The left and right of each box are the 25th and 75th percentile, and the centerline is the 50th percentile. We use a threshold of 500 molecules $cm^{-3} s^{-1}$ to partition fast and slow species. Panel (c) is the network of species blocks. A connection means that at least two species from these two blocks appear in the same reaction. The distance between the two blocks is proportional to the block distance as defined by Eq. 3.

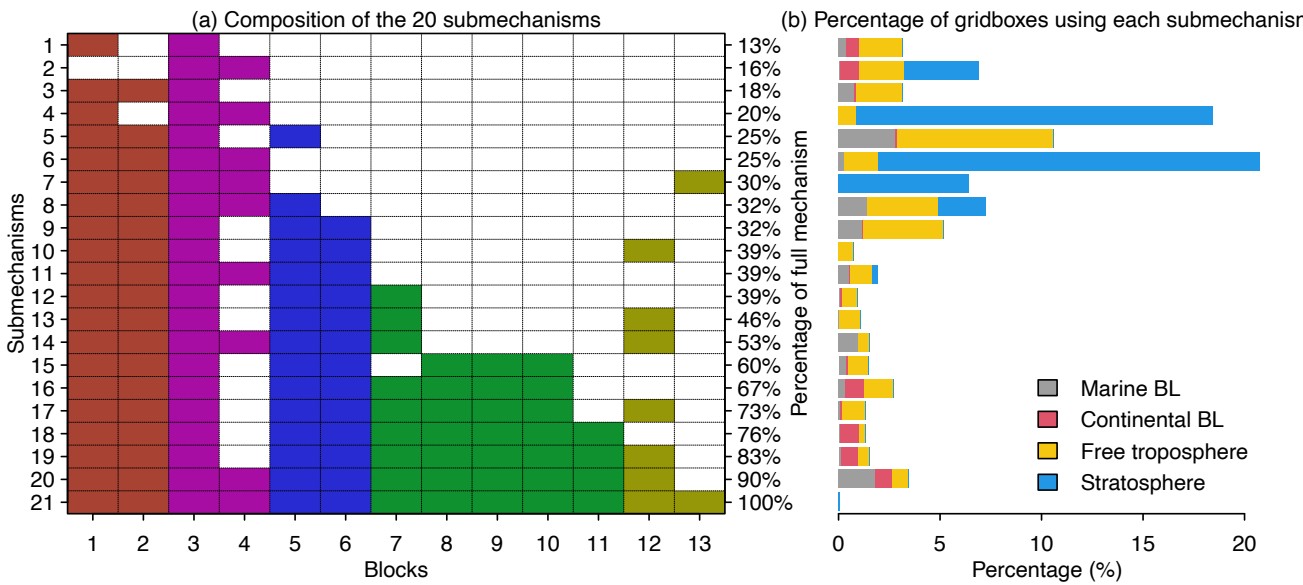

**Figure 5. Submechanisms and percentage of gridboxes using each mechanism.** Panel (a) shows the composition of the 20 submechanisms and full mechanism (the 21$^{st}$ one) as well as the percentage of species from the full mechanism that are treated as fast in each of them. Colors denote species block types as defined in Figure 4. Panel (b) shows the percentage of gridboxes using each submechanism in the marine boundary layer (BL, 0-2 km altitude), continental BL, free troposphere (2km to tropopause), and stratosphere.

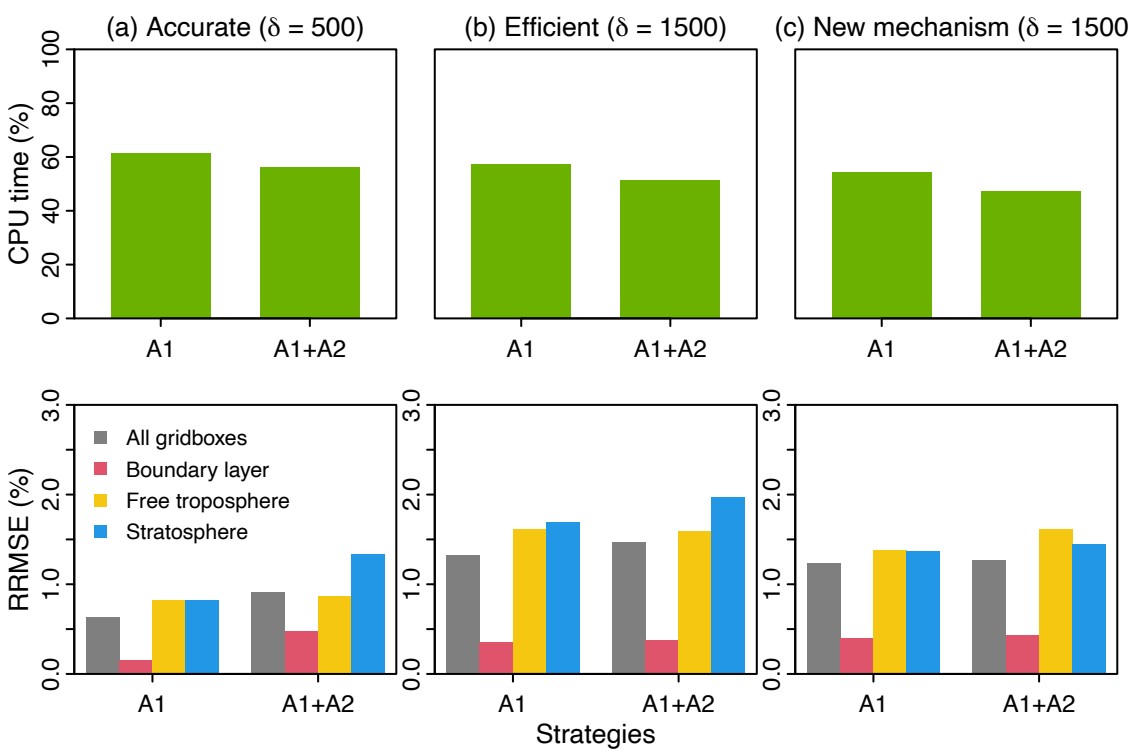

**Figure 6. Performance and accuracy of the adaptive chemical mechanism**. We test the performance of the adaptive method by (A1) removing slow species ($P_i$ or $L_i > \delta$) and (A2) removing slow reactions (reaction rate < 10 molecules cm$^{-3}$ s$^{-1}$). Results are shown on the last day of 3-year simulations. The unit of $\delta$ is molecules cm$^{-3}$ s$^{-1}$. The performance is measured by the computing processor unit (CPU) time used by the chemical operator, and the accuracy is measured by the median relative root mean square error (RRMSE) for species concentrations using the full chemical mechanism for all gridboxes, in the boundary layer (0-2 km altitude), free troposphere (2 km to tropopause), and stratosphere. For (a) and (b), we use e $\delta$ as 500 and 1500 molecules cm$^{-3}$ s$^{-1}$ in GEOS-Chem 12.0.0 that has 228 species and 724 reactions. For (c), we port the algorithm to GEOS-Chem 12.9.1 that has 262 species and 850 reactions. The number of blocks ($N$) is 13 and the number of chemical regimes is 21 (20 submechanisms (M=20) and one full mechanism).

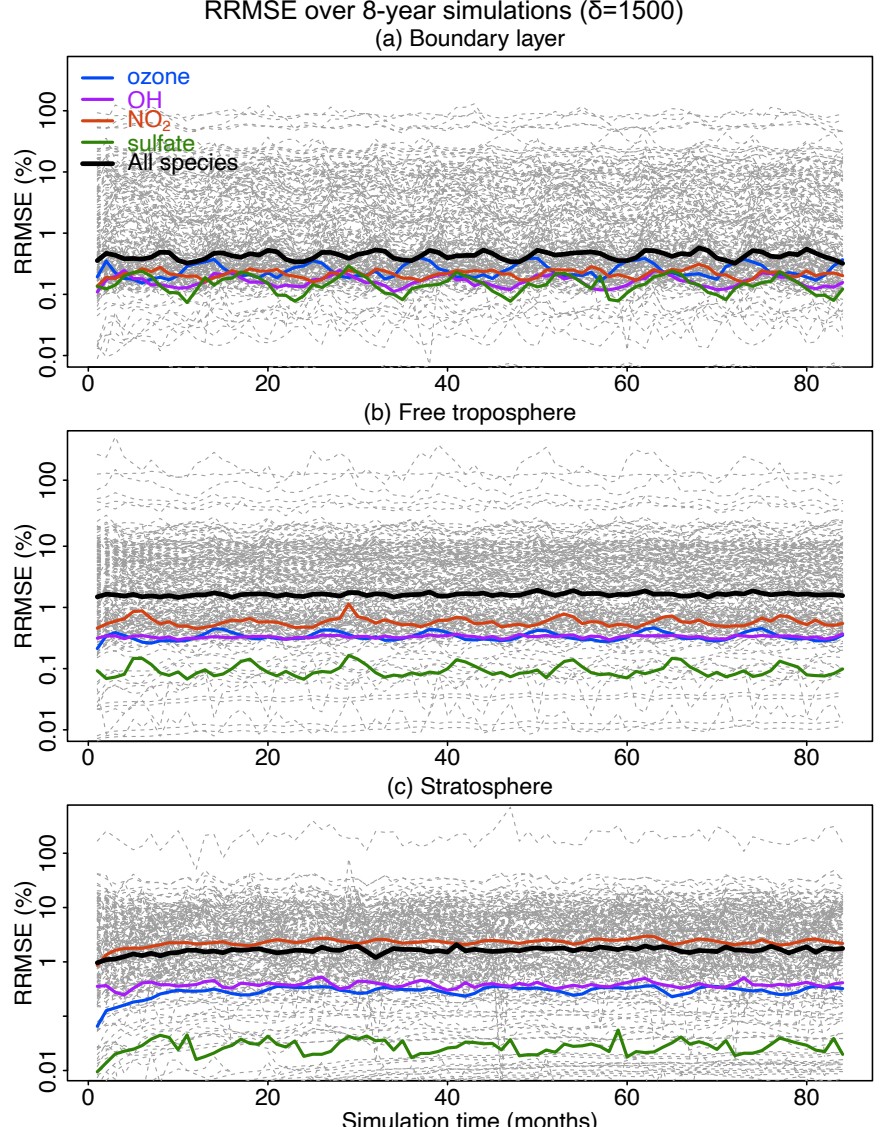

**Figure 7**. Accuracy of the adaptive reduced chemistry mechanism algorithm over an 8-year GEOS-Chem simulation using a threshold δ of 1500 molecules $cm^{-3}\,s^{-1}$ to separate fast and slow species. We show the RRMSE in the (a) boundary layer, (b) free troposphere, and (c) stratosphere. Results are also shown for the median RRMSE across all species in the mechanism and more specifically the RRMSE for ozone, OH, $NO_2$, and sulfate.

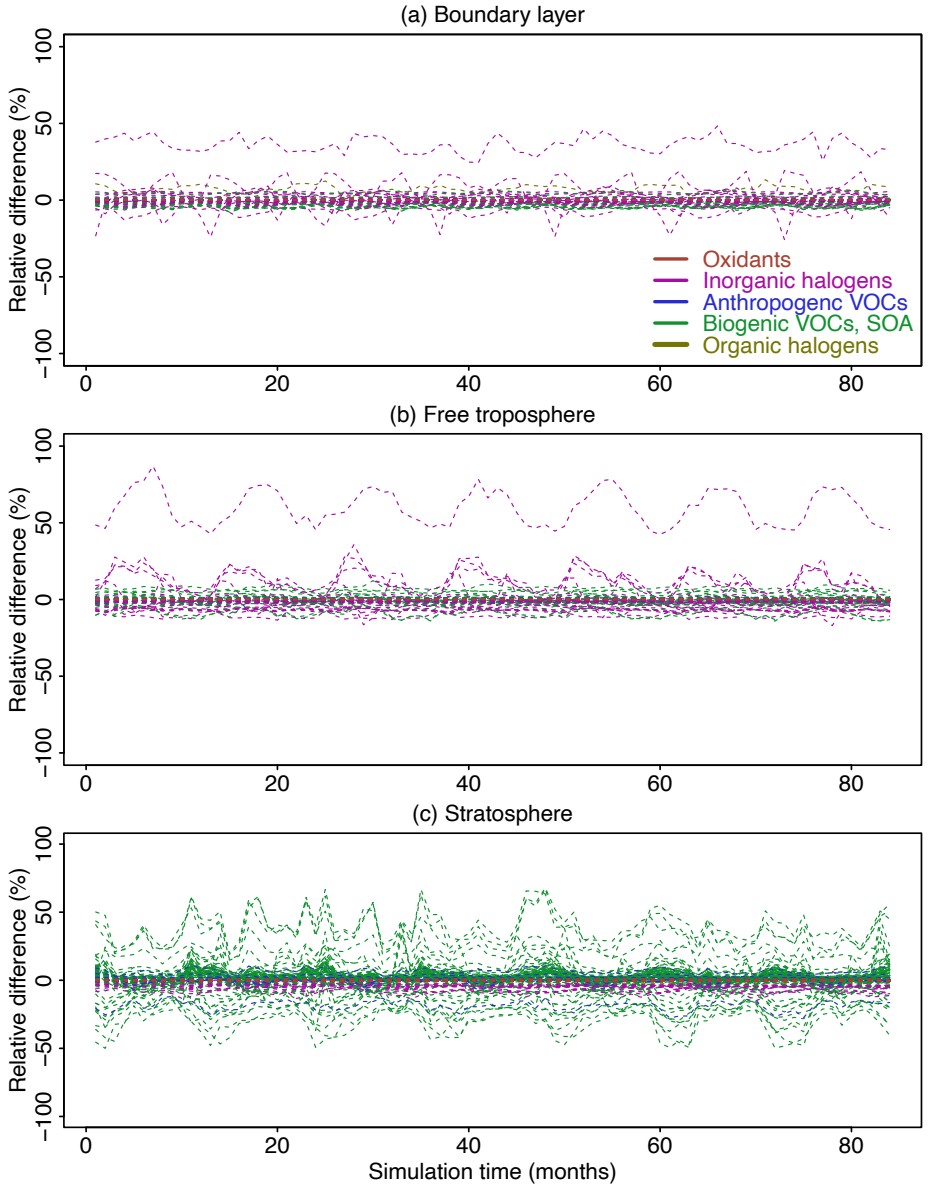

**Figure 8**. Relative difference of atmospheric masses in the adaptive reduced chemistry mechanism algorithm over an 8-year
450    GEOS-Chem simulation using a threshold $\delta$ of 1500 molecules $cm^{-3}\,s^{-1}$ to separate fast and slow species. We show the
RRMSE in the (a) boundary layer, (b) free troposphere, and (c) stratosphere. Different colours denote different species
categories (more details can be found in Table 1). Figure S8 presents more detailed results for the species with RRMSEs in
the -1.5% to 1.5% range.