# Peer review of "A machine learning-guided adaptive algorithm to reduce the computational cost of integrating kinetics in global atmospheric chemistry models: application to GEOS-Chem versions 12.0.0 and 12.9.1"

_Geoscientific Model Development, 2020_

## Author Comment (AC1)

**Response to referee comments on "A machine learning-guided adaptive algorithm to reduce the computational cost of atmospheric chemistry in Earth System models: application to GEOS-Chem versions 12.0.0 and v12.9.1"**

We thank the referees for their careful reading of the manuscript and the valuable comments. This document is organized as follows: the Referee's comments are in *italic*, our responses are in plain text, and all the revisions in the manuscript are shown in blue. **Boldface blue text** denotes text written in direct response to the Referee's comments. The line numbers in this document refer to the updated manuscript with tracked changes.

**Referee 1**

*The computational cost of atmospheric chemistry with atmospheric chemistry transport or an Earth System Models is large and methods to reduce these costs are so useful. The authors have over the years presented a series of papers (Santillana et al., 2010; Santillana et al., 2016; Shen et al., 2020) that attempt to reduce this computational burden by means of (simplifying here) separating the chemistry into fast species for which the differential equations need to be explicitly solved and slow species which can be solved analytically. The complexity of the approach has increased over the years and this paper represents the current incarnation of the methodology.*

*These technical advances in the numerical methods for atmospheric chemistry transport model (and more generally for geophysical models) are not seen as being sexy science, however, they are essential if these models are to be useful for the wider community. I am thus supportive of this paper and would suggest publication after some clarifications and corrections suggested below.*

*Major comments.*

***The figures from the supplementary material should be included in the main text of the paper.** For me, it is hard to understand some of the more complex mathematical aspects of the paper without a diagram to support it (e.g. S1 etc). I don't really see why all of the figures from the supplementary material can't be included in the main text. They are not particularly repetitive and I think it would be useful to the reader to see them all.*

**Response**. Thanks. We have moved Text S1, Figure S1, S2 and S6 to the main text.

***It would be useful at the end of the introduction to provide a context for what is coming up in the rest of the paper**. The lines around 185 provide this commentary about what has been done in the past, the problems associated with those and the method of addressing those which is discussed in the rest of the paper. This would help to contextualize Section 2 which seems like a rather remote set of definitions at the moment. In a few places, the paper feels like it is rather disjointed with one section not necessarily rolling into the next with much cohesion. Perhaps a re-read and a re-think of some of the structure would be beneficial for much of the paper.*

**Response**. Thanks. We have added a new paragraph at the end of the introduction to discuss what we have done in previous work and what to do next.

Line 69. In this work, we continue developing the adaptive method described by Shen et al. (2020). This method pre-assembles a small number of subsets of the full chemical mechanism representing the range of conditions in the troposphere and stratosphere, and selects the most appropriate submechanism to use in the model locally and on the fly. The submechanisms are constructed by first splitting the full mechanism's atmospheric species into $N$ different blocks based on similarity of chemical behaviors, using a machine

learning clustering method. We then define the submechanisms as different assemblages of blocks, select *M* of these assemblages to encompass the majority of chemical conditions in the atmosphere, and build them into the model. The choice of submechanism in the model is then made locally by computing chemical production and loss rates of the mechanism species and deciding which need to be part of the coupled chemical computation ('fast' species) and which can be tracked independently ('slow' species). A major development here is to define chemically coherent blocks that allow the method to easily accommodate changes in the chemical mechanism and to be readily applied to different mechanisms. We further improve the performance of the method by reducing the number of reactions as well as the number of species in the submechanisms.

*The available code is included in a RAR format. This doesn't seem to decompress on my Mac. Can we have the data in a standard zip format?*

**Response**. It can be decompressed after you remove .rar. I have clarified this in README in the same directory. Thanks.

***Minor Comments.***

*Line 60. This sounds like the end of an abstract rather than an introduction. At this point, the methodology hasn't been explained or tested so how can it be called chemically coherent, accurate or*

**Response**. Thanks. We now rewrite this sentence.

Line 77. A major development here is to define chemically coherent blocks that allow the method to easily accommodate changes in the chemical mechanism and to be readily applied to different mechanisms. We further improve the performance of the method by reducing the number of reactions as well as the number of species in the submechanisms.

*Line 87. Can the value of 10 molecules cm-3 s-1 be put into some context? Why was this chosen? Which reactions fall into this category?*

**Response**. Thanks. Now we say
Line 192. In each submechanism, **if a reaction is slower than 10 molecules $cm^{-3}$ $s^{-1}$ over all gridboxes that select this submechanism, this reaction is considered as unimportant in contributing to the threshold $\delta$ and is removed from the submechanism; but this reaction will be kept if it is faster than 10 molecules $cm^{-3}$ $s^{-1}$ in any of these gridboxes selecting this submechanism. The threshold we used to separate fast and slow reactions is slightly larger than 0 molecules $cm^{-3}$ $s^{-1}$ because of numerical precisions (unimportant reactions may still have a reaction rate > 0 molecules $cm^{-3}$ $s^{-1}$ in the numerical chemical solver in some timesteps)**. About 40-60% reactions can be removed using this strategy. For example, reactions of short-lived volatile organic compounds (VOCs) are removed in stratospheric gridboxes, and daytime photochemical reactions are removed in nightime gridboxes.

*Line 99. Giving the number of species (228?) in the reaction mechanism would be useful for contextualizing the 3400 other numbers?*

**Response**. Thanks. Now we say
Line 216. For example, in the reaction A+B-→C, there are 2 pairs (A-C and B-C) of reactants-products, 3 vertices (A, B, and C) and 2 edges (A-C and B-C).

*Line 104. "$T_{i,j}$ is the is the number of reactions that include both species i and j." Is that tas a reactant, products or either?*

**Response**. Now we say.

Line 285. Where $T_{i,j}$ is the number of reactions that include both species i and j (one is the reactant and the other is the product),

*Figure S1 should appear here to help explain the definition of Di,j. It would be useful to give an explanation for the numbers which are obtained given the chemical mechanism linking Toluene, xylene and Glyxoyl.*

**Response**. We now move Fig. S1 back to the main text. We have made it clear about how we calculate the numbers in the figure caption.

*The methods used to calculate "distance" appears to refer to the mechanism without any chemistry occurring. In the case of A+B-->C and A+D-->C, the "distance" between A and C doesn't care about the 2 rate constants or the concentration of the reactants. It would appear that chemically minor routes or channels are given the same weight as the dominant routes or channels. Presumably, the ideal way of working out the distance between species would be to explore the mechanism with some chemistry occurring and the weigh the distances between vertices by the flux between species or something like that? This obviously has a significant downside of being much more complex to implements and contextual (the distances would change with the chemical environment). It might be useful to describe this as being an optimal approach, but the approach taken is a simplification of this. Otherwise, it is rather hard to understand why the fluxes have not been used to represent the distance between species?*

**Response**. Thanks for raising this good point. We indeed considered weighting the distances using the logarithms of global average chemical reaction rates. However, we don't find this will make a big difference in our results but it makes the method very complex. Now we say this in text.

Line 299. We also tried weighting the species distances using the logarithms of their global mean reactions rates but this does not have significant effects on our final results.

*Line 116. It might be worth including the definition of "long-lived" and fast here.*

**Response**. Thanks. This sentence is removed in the revision and we have defined "long-lived" and fast elsewhere.

*Line 130. It would be worth pulling in the description of f from the supplementary information.*

**Response**. Now we move Text S1 back into the manuscript.

*Line 134. Bring in the figure S2 into the main body.*

**Response**. Done, thanks.

*Line 144. By only considering values greater than 1e6 cm-3 from your statistical analysis you are not excluding many values for typically high concentration species (O3 etc) but you will be excluding a significant fraction of the values for OH shown in Figure S6. It is not appropriate to do this for OH. A significant fraction the grid boxes will have OH concentrations less than 1e6 (the canonical global mean value). I appreciate that there might be increased errors at lower concentrations but if a large number of the global grid boxes have lower concentrations than this, this metrics is being rather selective in the grid boxes that it is using the analyse the*

*model. From Figure S8 it would appear that this excludes evaluation of the model error in large chunks of the surface of the globe for OH?*

**Response**. We now test also other lower thresholds ($10^5$). See Figure S5-S6 for more details.

[Figure]

**Fig. S5**. Relative error from the adaptive mechanism reduction method after three years of simulation in the GEOS-Chem global 3-D model for tropospheric-stratospheric chemistry. The figure shows relative differences of 24-h average OH, ozone, sulfate and $NO_2$ concentrations relative to the full-chemistry simulation on the last day of the three-year simulation. The calculation removes slow (P and L < 1500 molecules cm$^{-3}$ s$^{-1}$) and slow reactions (rate < 10 molecules cm$^{-3}$ s$^{-1}$). The number of blocks ($N$) is 13 and the number of chemical regimes ($M$) is 20. We only consider gridboxes with species concentrations >= $10^6$ molecules cm$^{-3}$; otherwise the gridboxes are shown as white.

[Figure]

**Fig. S6**. Same as Fig. S5 except that we only consider gridboxes with species concentrations $\geq 10^5$ molecules cm$^{-3}$.

*It's also not clear why the value is set at 1e7 for NO2 in Figure S6? Is this alternative cut off used in Figure S8? Presumably, this is the reason why the polar regions are excluded from Figure S6 for NO2?*

**Response**. Now we use the same thresholds for all species and we also test lower thresholds (a=$10^5$). See Figure S5 and S6 for more details

*Line 158. Can Figure 1 include information about the definition of slow and long-lived for reader clarity otherwise they are having to flick back to find the definitions used?*

**Response**. Done, thanks.

*Line 182. "different blocks based on similarity of chemical behaviours using a machine learning clustering method." I think this is described in the supplementary material. This should be brought into the main text of the paper or a reference to where this is described included in the text.*

**Response**. Thanks, we have moved this part back to main text.

*Line 185. It would be very useful to have had the contextual information given here much earlier in the paper so the reader can understand what didn't work in the past, what the proposed solution is and how this will be implemented.*

**Response**. We now move this to the last paragraph of the introduction part. Please check there for more details.

*Line 209. I'm not sure I understand the comment that iodine reservoirs are inert? Many of them are highly photolabile. This doesn't seem to make sense?*

**Response**. I think the more active iodine reservoirs are in Block 3. The species in Block 4 are relatively inert and they are considered as fast in only <5% of gridboxes.

*Line 243. When talking about the error this is the RRME? This should probably be clarified. It would also be useful to discuss the implications of the >1e6 cm-3 limit here on the statistics. Is the NO2 value with the 1e7 cm-3 limit as indicated in the supplementary material?*

**Response**. We use a different threshold $NO_2$ just for visualization in the old manuscript and we use $10^6$ molecules cm$^{-3}$ s$^{-1}$ for all species now. See Figure S5-S6 for more details.

And we have clarified that whether we use the RRMSE error or the relative difference compared to the standard simulation in the text.

Now we say this in text.

Line 721. Computing the RRMSE for all species with concentrations higher than a=$1\times10^5$ molecules cm$^{-3}$ (instead of $1\times10^6$ molecules cm$^{-3}$) shows similar results except that the magnitude of the error is higher because the relative difference is expected to be higher at low species concentrations (Fig. S6, S8, S9).

*Line 245. Being able to update the chemical mechanism is an important aspect of maintaining the viability of the model in the long term. If there was a significant change to the mechanism the whole process of running the model without the chemical mechanism splitting would need to be done again? The method outlined here is for "on the fly" updates and relies upon the changes being small and the chemical intuition of the person doing the update. It would be useful to explain that the approach described here is for "patching" the mechanism etc. rather than updating the whole mechanism*

**Response**. Thanks for pointing this out. Now we make it clear how to update our mechanism.

Line 725. Chemical mechanisms in models are frequently updated, including addition and removal of species. Because the species blocks are chemically coherent, our algorithm can accommodate mechanism updates without requiring reconstruction of the submechanisms. New species simply need to be added to the appropriate blocks. Figure S10 shows the diagram for adding new species into the mechanism. Attribution of a species to a given block can be easily determined by its chemical family and the percentage of gridboxes that treat this species as fast when averaged globally. In order not to compromise the computational efficiency, the basic rule is to not mix faster species with slower ones.

*It's not clear where new "biogenic VOC" degradation products would go (blocks 7,8 or 9 etc). If an exact mechanism for working out the placement hasn't been found and new species were randomly allocated to 7,8 or 9 (or another mechanism) it would be useful to have that documented.*

**Response**. Thanks. Now we say this in text.

Line 729. In order not to compromise the computational efficiency, the basic rule is to not mix faster species with slower ones. For example, biogenic VOC species and their products could go to Block 8-9 if the percentage of gridboxes that treat them as fast is >1% or Block 10-11 if the percentage is <1%. Our algorithm is robust to misplacements of new species, which may affect computational performance but will not enlarge the error.

*Line 272. I don't think that it has been demonstrated that it can be ported to different atmospheric models "easily". Relatively minor changes to the chemical mechanism within one model were shown to be able to be incorporated without re-running the whole tuning procedure but I don't think it can be demonstrated that it is easy to move into a different model (CESM etc).*

**Response**. Thanks, and we have deleted this sentence.

*Figure 3. Could be greyed out area showing the number of grid boxes in that category be split into some subcategories: marine boundary layer, continental boundary layer, free troposphere, stratosphere etc to provide some additional information?*

**Response**. Done.

[Figure]

**Figure 5**. **Submechanisms and percentage of gridboxes using each mechanism**. Panel (a) shows the composition of the 20 submechanisms and full mechanism (the 21ˢᵗ one) as well as the percentage of species from the full mechanism that are treated as fast in each of them. Colors denote species block types as defined in Figure 4. Panel (b) shows the percentage of gridboxes using each submechanism in the marine boundary layer (BL), continental BL, free troposphere, and stratosphere.

*Figure 4. This is the performance over what timescale? All of the 1-year timesteps for all grid boxes? It's not clear that the bars indicate the simulation speed up and the symbols represent the accuracy.*

**Response**. They are based on the last day of 3-year simulations and we have made this clear in the figure caption.

*Supplementary material. Much if not all of this should be in the main body of the paper.*

**Response**. Thanks. We have moved Figure S1, S2 and S6 to the main text.

*Figure S4. Is this mislabelled as "anthropogenic blocks", think it should be biogenic blocks? How is the decision made about fast/slow when there are multiple blocks? This could be explained.*

**Response**. Done, thanks.

*Figure S5. I'm not sure that the figure actually shows the "mechanism complexity needed"? It shows the sub-mechanism at each location. I found the % fast labelling slightly confusing as it wasn't that obvious whether the colour scale was describing the Regime or the % fast? Perhaps the % fast could just be removed for simplicity?*

**Response**. Now we say this the caption.

**Fig. S3.** Chemical mechanism complexity used in the adaptive chemical mechanism in different regions of the atmosphere.

We prefer to keeping the "%fast" because it can better visualize the complexity of each chemical mechanism.

*Figure S7. How do these curves look if the <1e6 cm-3 restriction in calculating the RRMS is removed?*

**Response**. The relative error will become very large for species with low concentrations. So here we prefer to show the curve only gridboxes with enough high species concentrations. But we do include a new figure in the supplementary by considering all gridboxes with species concentrations >1e5 cm$^{-3}$.

[Figure]

**Fig. S9**. Same as Fig. S4 except that we consider gridboxes with species concentrations >= $10^5$ molecules cm$^{-3}$.

*Figure S8. Can the location where the RRMS has not been calculated due to the 1e6 cm-3 restriction be indicated (grey the area?).*

**Response**. Done, please check Fig. S5 and S6 for more details.

*Figure S9. What is the H / L notation indicating?*

**Response**. Now we make this clear in the caption.

Fig. S10. 'H' ('L') means higher (lower) percentage of gridboxes that consider this species as fast.

---

## Author Comment (AC2)

**Response to referee comments on "A machine learning-guided adaptive algorithm to reduce the computational cost of atmospheric chemistry in Earth System models: application to GEOS-Chem versions 12.0.0 and v12.9.1"**

We thank the referees for their careful reading of the manuscript and the valuable comments. This document is organized as follows: the Referee's comments are in *italic*, our responses are in plain text, and all the revisions in the manuscript are shown in blue. **Boldface blue text** denotes text written in direct response to the Referee's comments. The line numbers in this document refer to the updated manuscript with tracked changes.

**Referee 2**

*This paper present some clearly very well thought out and well executed methodologies that are shown to be effective at minimising computational costs for solving a complex chemical mechanism in the GEOS-Chem model. These tool are likely to have substantial benefits for the air quality and Earth System modelling communities. Unfortunately, the writing of the paper is confusing at times, understandable given the complexity of the subject matter but it does distracts from messages it is trying to convey. With a few small changes to the structure and presentation, this will be an excellent paper well suited for publication in GMD.*

**Major Comments**

*The biggest issue I find with the paper is the it goes into the nitty gritty of how it partitions different species/reactions into different categories (sections 2.2-2.5) before it explains the overall design philosophy and what these different categories are used for (in section 3.1). The result is that on first reading, I got very confused reading through section 2 and was only able to make sense of it on second reading. This could be improved substatially if either Section 2.1 were expanded upon to give an overview of the whole design philosophy and what each of the the different subcategories (fast, slow species, unimportant reactions etc.) are going to be used for before they are described in detail, or a new Section 2.2. is added giving an overview of all of the developments. Some of this description is in the first paragraph of Section 3.1. Also section 3.1 repeats a lot of the text currently in section 2.1 about the model description and is inapropriate here, therefore section 3.1 should also be editted to avoid repition. In short, Section 2 should contain all of the descriptions and definitions for the model and the adaptive algorithm for the chemical operator. Section 3 should focus on testing, evaluating and optimising the algorhithm in the 3D model. If these two sections were more clearly defined, the paper would be much easier to follow.*

**Response**. Thanks. We have made the following changes to improve the presentation quality.
1. We now give an overview of the whole design philosophy in the last paragraph of the introduction part (as also suggested by the first referee).
2. We have moved part of the text in the first paragraph of Section 3.1 to Section 2.1 and 2.2.
3. We have moved Text S1 back to Section 2 to give a better description of our method.
4. We have moved Figure S1, S2 and S6 to the main text.

*There are also some inconsistancies in language and definitions - I found the use of "slow species" and "slow reactions" (each of which use different threshold definitions) particularly confusing. It would help to be consistent and use "unimportant reactions" for those <10 molecules cm-3 s-1. I have specific comments below to help with this.*

**Response**. Thanks. We now use 'slow reactions' throughout the paper because it is more accurate than 'unimportant reactions'.

*In terms of the statistical analysis (Section 2.6) it is important to have a measure of bias as well as error - I would generally be more concerned about a species that has an error of 1% and bias of ~1% (which would imply a consistant error in one direction), than one which has an error of 1% and a bias of ~0% (which would implpy more random error). Looking at Figure S8, the errors in surface Ozone seem to be biased in one direction. That is potentially concerning if the bias is enough to significantly affect tropospheric ozone burden - as ozone is a radiatively important species this could have consequences in an Earth System model. Please also include a normalised measure of bias in Section 2.6.*

**Response**. Thanks for this good point. We now define a relative abundance metric in the paper.

Line 465. A second metric to evaluate our adaptive chemical mechanism is the relative difference of atmospheric abundances for all species compared to the standard simulation. This tests for accumulating bias over long simulation periods.

The tropospheric ozone doesn't show accumulated bias in one direction. As seen from Figure 7, its bias has some seasonality. And now we have a new figure to show the relative difference in atmospheric abundances of all species.
Line 719. The median relative difference in atmospheric abundances among all species remains at 0% over this 3-year period; the relative differences for key species like ozone, OH, sulfate and NO2 also remain at 0% and are within ±10% for >99% of the other species (Figure S7).

[Figure]

**Fig. S7**. Relative difference of total atmospheric abundances using adaptive reduced chemistry mechanism algorithm over a three-year GEOS-Chem simulation. Results are shown for the median relative difference across all species in the mechanism and more specifically for ozone, OH, NO2, and sulfate. The grey dashed lines are for all 228 species.

*Finally, there are a number of figures in the supplement which would have been useful in the main paper and I did not understand why they were not in the main paper. I would recomend moving at least figures S1, S2, S5, S6, S7 and S8 into the main paper.*
**Response**. Thanks. After considering both referee's suggestions, we now move Figure S1, S2 and S6 back to the main text.

**Specific comments**

*ln 26. Ambigous "it" in: "because it exerts strong forcing and feedbacks...". change to "because chemical and aerosol species extert strong forcings and feedbacks..."*
**Response**. Done, thanks.

*ln 49. Change "guide us build" to "guide us **to** build".*
**Response**. Done, thanks.

*Ln 88. here and elsewhere, please be consistent and use the term "unimportant" instead of "slow" to describe those reactions that are removed with fluxes <10 molecules cm-3 s-1, otherwise it is easily confused with the "slow species".*
**Response**. Thanks. But we think 'slow reactions' are more accurate than 'unimportant reactions', so we use it throughout the paper.

*Section 2.3. If I am to understand this correctly, when species are defined as being "slow" and/or "long-lived", they are not "removed" from the mechanism per se, rather they are solved using the analytical approach instead of the 4-th order Rosenbrock solver. Given this, I think you need to make clear here that when you say "coupled system", you mean all of the species and reactions which are solved using the Rosenbrock solver. All of the species that are "removed" or "excluded" from the coupled system, as you say later, are instead solved using the simple analytical approach.*
**Response**. Thanks for correcting this. Now we say
Line 203. We solve the fast species in their submechanism using the standard Rosenbrock solver. For the slow or long-lived species, we approximate the evolution of concentrations using an explicit analytical solution that assumes first-order loss (Santillana et al., 2010).
Line 209. **As such, we still update the concentrations of all species but in a more efficient way**.

*Line 99. Please can you define "distance" qualitively here before the quantitative definition.*
Response. Now we say
Line 212. We construct coherent subsets ('blocks') of the species in the mechanism species based on their linkages through the mechanism reactions. This is done objectively by defining the species distances in the mechanism using graph theory. **In general, two species should have shorter distances if they appear in the same reaction more times and have similar products in the mechanism**.

*Line 112. The term "distance" is overloaded in the text to mean multiple different things. You have defined a new term "Euclidian distance" |Di-Dj|, which is a scaler, rather than the previous "distance" Di which is a vector. This Euclidian distance is then modified to cluster species together (I would call this something like "clustered distance"). Please make clear that it is this "clustered distance" that is stored in the matrix and used to calculate the cost function Z (eq 4).*
**Response**. Thanks, we prefer to name it as 'modified distances'. Now we say this in the text.
Line 298. We store these modified distances of all pairs in a 228x228 matrix.

*Line 113. When applying the 50% factor, is this applied iteratively? i.e. if two species are each others closest pair, will their Euclidian distance be reduced by a factor 0.5\*0.5=0.25, as iterating through each species, or is the 50% factor only applied once? Please be clear which approach you are using.*
**Response**. This scaling will be applied only once.
Line 297. Second, for each species $i$, we will decrease its distance with the 5 species that have highest similarity with it by 50% and **this scaling is applied once for one pair of species**.

*The 50% factor and 5 closest neighbours both seem quite arbitrary, but I think I can see how this approach will cause clustering of species into close families. What was the reason for the use of these two values, and did you test other values?*
**Response**. We indeed tested other numbers and the results are in general consistent.
Line 297. Using 10 highest-similarity species instead of 5 and decreasing distances by 30% or 70% does not change the results.

*Line 135. Would benefit from Figure S2 being in the main text here. Looking at Figure S2, I think you can make a fair cost-benefit argument that M=20, N=13 is well optimised as it is on the bottom-left of the 30% contour. It is clear from the contour lines that you get diminishing returns of the fraction of species if you were to increase M and N, hence selecting on the 30% contour seems reasonable. By selecting the bottom-left part of the 30% contour, you are minising both N and M.*
**Response**. Thanks, we now move Figure S2 back to the main text. Now we say this in text.

Line 528. In order to make the code manageable, we choose to use $M = 20$ resulting in an optimal value $N = 13$ at which only 30% of the species need to be treated as fast in the global tropospheric and stratospheric domain (Figure 3).

*Basically, you can be more rigorous in the justification for why you used M=20, N=13 than simply saying you "choose" them.*
**Response**. Now we say this in the main text.
Line 528. In order to make the code manageable, we choose to use $M = 20$ and an optimal value $N = 13$ at which only 30% of the species need to be treated as fast in the global tropospheric and stratospheric domain.

*Line 137-139. Please move this text to the top of section 2.5, as this describes the training set used to derive the submechanisms and blocks.*
**Response**. Done, thanks.

*Line 196. Please again change "slow reactions" to "unimportant reactions". Please clarrify - are the unimportant reactions removed from each of the submechanisms using the original training data, or are they removed on the fly depending on the concentrations of species in each grid cell at each timestep?*
**Response**. Thanks. We prefer to name it 'slow reactions' and we have made it clear that the slow reactions removed in each submechanism are pre-defined and the same.
Line 524. The slow reactions removed in each submechanism are pre-defined (see Section 2.2 for more details).

*Removing the reactions from each of the submechanisms in advance seems like the more efficient approach to me. However, there is a risk that the approach becomes inconsistant if used in different time periods with different chemical conditions to the training data. For example, there will be risk in using submechanisms derived with present day training data in preindustrial conditions. This is relevant for application in Earth System models.*
**Response**. Thanks for raising this point and we now discuss it in the text.
Line 200. Here we remove these slow reactions in each submechanism based on present-day atmospheric chemistry environment and it should be re-evaluated if this method is applied in other periods (e.g. pre-industrial times) when the atmospheric conditions could be very different from our present-day one.

*line 229. The full mechanism is by definition not a submechanism. Say that it is the 21st "chemical regime".*
**Response**. Done, thanks.

*Line 236. slow reactions -> unimportant reactions*
**Response**. Now we use 'slow reactions' throughout this paper.

*Figure 1. The line in panel a shows the fraction of species removed from the* coupled mechanism. *Call the slow reactions unimportant reactions.*
**Response**. Now we use 'slow reactions' throughout this paper.

*Figure 3. There are 21 chemical regimes, made up of M=20 submechanisms plus the whole mechanism.*
**Response**. Now we use 'slow reactions' throughout this paper.

*Figure 4. Number of chemical regimes is 21 (M+1), not 20.*
**Response**. Done.

**Supplementary material**

*line 12. think $Z_2$ should be called f or f(M,N) instead. Use a different term to D for the gridcells, because D is already used for the distance vectors.*
**Response**. Fixed and we now use P instead of D.

*line 18-20. Unclear how this algorithm works, don't know what is meant by "temperature" here.*
**Response**. Now we add a reference to describe how this temperature parameter works.

*figure S5. presumably chemical regimes should go up to 21, not 20? Also, its the 21st regime that has 100% of the mechanism, regime 20 has 90% according to Figure 3.*
**Response**. We have updated this figure. Please check Figure S3 for more details.

*Figure S6. Please include panels showing biases for the key species. If any have consistant/growing biases, that should be commented on (especially concerning for ozone). What value of δ did you use here?*
**Response**. Thanks for this good point. We now define a relative abundance metric in the paper.

Line 465. A second metric to evaluate our adaptive chemical mechanism is the relative difference of atmospheric abundances for all species compared to the standard simulation. This tests for accumulating bias over long simulation periods.

And now we have a new figure to show the relative difference in atmospheric abundances of all species.
Line 719. The median relative difference in atmospheric abundances among all species remains at 0% over this 3-year period; the relative differences for key species like ozone, OH, sulfate and NO2 also remain at 0% and are within ±10% for >99% of the other species (Figure S7).

[Figure]

**Fig. S7**. Relative difference of total atmospheric abundances using adaptive reduced chemistry mechanism algorithm over a three-year GEOS-Chem simulation. Results are shown for the median relative difference across all species in the mechanism and more specifically for ozone, OH, NO₂, and sulfate. The grey dashed lines are for all 228 species.

---

## Author Response (AR2)

**Response to referee comments on "A machine learning-guided adaptive algorithm to reduce the computational cost of atmospheric chemistry in Earth System models: application to GEOS-Chem versions 12.0.0 and v12.9.1"**

We thank the editor for his careful reading of the manuscript and the valuable comments. This document is organized as follows: the editor's comments are in *italic*, our responses are in plain text, and all the revisions in the manuscript are shown in blue. **Boldface blue text** denotes text written in direct response to the Referee's comments. The line numbers in this document refer to the updated document with tracked changes.

**We have made these major changes.**
1. We conducted an 8-year simulation from 2013 to 2020 and found that the error is still stable over this period.
2. We removed all texts claiming that our algorithm can be used in earth system models. See tracked changes in the manuscript for more details.
3. We have a new Figure 8. We show the error of all species in Figure 7, 8 and all related figures in the supplementary materials.
4. We calculate the errors using three different definitions as suggested by reviewers and the editor. See section 2.5 for more details.

[Figure]

**Figure 8**. Accuracy of the adaptive reduced chemistry mechanism algorithm over an 8-year GEOS-Chem simulation using a threshold δ of 1500 molecules cm$^{-3}$ s$^{-1}$ to separate fast and slow species. (a) Same as Figure 7c but for the 8-year simulation. Here we use an absolute threshold of species concentration ($a = 1 \times 10^6$ molecules cm$^{-3}$) to define the RRMSE (See Section 2.5). (b) Same as Figure 8a but using a relative threshold ($a = \max(5^{th}$ percentile, $1 \times 10^4$ molecules cm$^{-3}$)) to define the RRMSE (See Section 2.5). (c) Relative difference of global atmospheric masses for each species. Dashed lines show results for all 228 species in the mechanism. Results are also shown for the median RRMSE across all species in the mechanism and more specifically the RRMSE for ozone, OH, NO$_2$, and sulfate.

Major comments

*One of my main concerns is the superficial and nature of discussion you offer, often having little physical context. Such instance are:*

*Response to Reviewer 1 comment on Line 87: Your response still does not offer any context regarding the*

*selection criteria. Why 10 molecules and not 100 or 1000? How much larger is "slightly larger than 0 molecules cm-3 s-1" you mention? What "because of numerical precisions" means here at all? I cannot accept this response as scientific.*

**Response.** We examined different thresholds in some short-term experiments in the early stage of this project. We found that the error quickly increases if the threshold is above 10 molecules cm$^{-3}$ s$^{-1}$. You are right that our previous discussion was superficial, and we provide more details here.

Line 143. In each submechanism, if a reaction is slower than 10 molecules cm$^{-3}$ s$^{-1}$ for all gridboxes that select this submechanism, then the reaction is considered negligible and removed from the submechanism. **The logic is that such a slow reaction will not contribute significantly to the total species production/loss rate threshold δ =500-1500 molecules cm$^{-3}$ s$^{-1}$ for the species to be included in the Jacobian matrix.**

*Similarly,*
*Reviewer 1 on Line 243, Reviewer 2 on Line 297 (and generally comments/replies regarding the choice of thresholds): You select a new threshold without explicating why – an approach "we use now lower value and it does (or does not) change things" is vaguely related to the nature of the selection, it does not help understanding which factors are important or not. Where are the important thresholds for distances? Why testing for 10 species is now sufficient as compared to 5 species before? It appears to be an arbitrary choice.*

**Response.** Thanks for this good point. This was done with repeated tests of different thresholds and then we determine if the optimized chemical blocks are consistent with our chemical intuition. We rewrite it here.

Line 208. Second, for each species *i*, we decrease its distance with the 5 species that have highest similarity with it by 50% and this scaling is applied only once for each species pair. We found in tests that using 10 highest-similarity species instead of 5 and decreasing distances by 30%-70% instead of 50% did not significantly change the results. The logic is that the number of species with similar chemical characteristics is usually around 5-10 and decreasing the distances among them by 30-70% can increase the probability of these species to be in the same chemical blocks after the optimization process.

*Furthermore,*
*Reviewer 1 on Figure S6, Reviewer 2 major comment: Showing the average difference for the species is alike reporting the average body temperature for all patients in the hospital. Not only that does not help understanding the biases for the species of importance, how would you judge/set the input (e.g. weight) of every species to this average?*
**Response**. Thanks. Now we show the error of all species in Figure 6, 8, S4, S7 and S8. Please check them for more details.

*A large bias in simulating N2O in troposphere (where it is mostly inert) will hardly affect CH4 lifetime there, whilst a small bias in OH will do so (especially on the long-term, I will return to that below).*
**Response**. We don't simulate the evolution of long-lived species in our standard chemistry model, following standard practice of simulations focused on tropospheric chemistry – we now clarify that this is our focus. We have also shown that the error is stable in an 8-year simulation.

Line 113. In the simulations presented here, methane, N$_2$O, and other long-lived halocarbons have fixed concentrations in surface air (Eastham et al., 2014; Murray, 2016) so that the longest resolved chemical modes are less than a year.

We also removed all text claiming that our algorithm can be applied in earth system models because we haven't tested it. We have also changed the title,

Title. A machine learning-guided adaptive algorithm to reduce the computational cost of integrating kinetics in global atmospheric chemistry models: application to GEOS-Chem versions 12.0.0 and 12.9.1

*Fig. S7 and its vertical scale are deceptive – behind larger relative differences for some species (not indicated which) one does cannot check for growing biases (I spot one in the right panel for NO2 nonetheless)*

**Response**. Thanks for pointing this out. Now we show all errors (in log scale) of all species in figures. Please check updated Fig. 7, 8 and S7.

*The stated errors that "also remain at 0%" make no sense from the numerical point of view – does this mean that you compute numerically identical result (read copy it)? So how large are deviations really? Do you mean below 0.1%? The metric "+-10% for >99% of the other species" is also not reasonable – what if the remaining 1% of species has other important (in addition to key ones) species, e.g. reactive compounds?*

**Response**. Sorry, our old narrative wasn't precise, and we have removed all these sentences. We now show all errors in Figure 7, 8 and the distribution of errors in Figure S4.

Line 376. The relative differences of global atmospheric masses are within 10% for >99% species (Figure 8c) and show no sign of increasing trends. The relative difference for NO$_2$ increases slightly from 0% to 0.4% in the first 30 months and then stabilizes at 0.4%. Key species like OH, ozone and sulfate have a relative difference smaller than 0.01% throughout this simulation period.

*I give a few hints to help adequately re-address these comments. Classical ODE integrators employ absolute and relative tolerances for estimating the errors about the solution that is obtained. This is done to constrain errors when relative scales of changes for the spectrum of variables integrated vary many orders of magnitude, and is shown to be valid using through the strict mathematical analysis (see, e.g., [1], Chapter on ODE integration). Whilst I do not require such here, **a minimum sound mathematical analysis of the order of the errors in your method should be present in the revised version**.*

**Response.** Thanks, but we are afraid that a simplified ODE assumption may not be applied here. We mentioned in text that the Rosenbrock chemical solver is a 4$^{th}$-order implicit method and the linear approximation for slow species is a 1$^{st}$-order explicit method. But we feel it is difficult to further analyze the order of the errors mathematically because of operator splitting with other processes including deposition and transport.

*A shortcoming of your current method, analysis and presentation of the results is also in using only absolute thresholds. Ozone and OH differ by orders of magnitude, yet you use the same absolute threshold values for these two species in the analysis. Presenting errors at the surface and at 15 km altitude (Figs. S6-7) has little sense for OH, whose tropospheric sink is 80% in the free troposphere. Which atmospheric abundances are compared in Fig. S7, are these gridboxes or domain averages? These are only a few questions, I could go on… but they already show how inconclusive and not-well-thought-through are the analysis and discussion you present.*

**Response.** Thanks for these good points. Here we report the error using both absolute and relative thresholds. See Figure 8 for more details. And we have a new paragraph discussing this.

Line 272.

$$RRMSE_i = \sqrt{\frac{1}{Q_i} \sum_{j=1}^{Q_i} \left(\frac{n_{i,j}^{reduced} - n_{i,j}^{full}}{n_{i,j}^{full}}\right)^2} \quad \text{(where } n_{i,j}^{full} \geq a\text{)} \tag{7}$$

Absolute threshold $a = 1 \times 10^6$ molecules cm$^{-3}$

Relative threshold $a = \max (5^{th}$ percentile, $1 \times 10^4$ molecules cm$^{-3})$

Line 372. Our algorithm also shows no sign of increasing errors over an 8-year simulation. Figure 8a-b displays the RRMSE over this period by taking account of all gridboxes with species concentrations above an absolute threshold of $1 \times 10^6$ molecules cm$^{-3}$ or a relative threshold of the 5th percentile of species concentrations across all gridboxes (see Section 2.5 for more details). In both cases, the median and the maximum RRMSE remain constant over this simulation period. The relative differences of global atmospheric masses are within 10% for >99% species (Figure 8c) and show no sign of increasing trends. The relative difference for NO2 increases slightly from 0% to 0.4% in the first 30 months and then stabilizes at 0.4%. Key species like OH, ozone and sulfate have a relative difference smaller than 0.01% throughout this simulation period.

*To recap, you need to substantially improve the analysis of errors, presentation and discussion in order to confirm that your method is accurate and convergent (w.r.t. the reference) and support your statements, especially the ones in the conclusion section. Please add a definition or explicate what exactly you imply by "chemical coherence" (better in the introduction).*

**Response**. Thanks for pointing this out. We use chemical initiation instead.

Line 69. A major development here is to enforce that chemically connected species be grouped in the same blocks, so that the blocks are consistent with chemical intuition and can be logically modified and extended as the mechanism changes.

*As of now, you show no evidence of "no error growth over multi-year global simulations" because many atmospheric compounds have lifetimes longer than 3 years (read they need to equilibrate to be consistently compared to the reference). Fig. S7 adds dubiety here, too. I suggest that you simulate at least a 30 years long period in order to corroborate that no substantial errors in CH4, NOx and other reactive N reservoirs and hence O3 are building up in the optimised mechanism.*

**Response.** A 30-year simulation is not necessary for tropospheric oxidant/aerosol chemistry, which we now clarify as our focus. Here we conduct an 8-year simulation and we find the error is stable over this time period. Please check Figure 8 for more details.

Line 372. Our algorithm also shows no sign of increasing errors over an 8-year simulation. Figure 8a-b displays the RRMSE over this period by taking account of all gridboxes with species concentrations above an absolute threshold of $1 \times 10^6$ molecules cm$^{-3}$ or a relative threshold of the 5$^{th}$ percentile of species concentrations across all gridboxes (see Section 2.5 for more details). In both cases, the median and the maximum RRMSE remain constant over this simulation period.

Note that an error of a few percent in oxidising radicals may lead to largely diverging results in a few decades. For instance, using a simple spreadsheet integration you may infer that a longer by 1% (as compared to assumed average of 9 years) atmospheric CH4 lifetime would yield CH4 abundance larger by already 3% (above 50 ppb, or growth rate in last 5 years!) in just recent 30 years, assuming that emissions are the same in both cases. So can your optimised method really simulate 3 decades of CH4 evolution with an accuracy of 1%?

**Response**. We don't simulate the evolution of long-lived species in our standard chemistry model. We now make it clear in text.

Line 114. In the simulations presented here, methane, N₂O, and other long-lived halocarbons have fixed concentrations in surface air (Eastham et al., 2014; Murray, 2016) so that the longest resolved chemical modes are less than a year.

Minor comments

*Reviewer 1 on Line 212: What are the objective criteria, if you state "This is done objectively here…".*
**Response**. We removed 'objectively' and now explain in detail how we construct coherent blocks in the following sentences.

*How can "two species … appear in the same reaction more times"? Present an example.*
**Response**. "appear in the same reaction" was confusing. We have replaced by "appear together in multiple reactions'. For example, NO and NO2, OH and HO2 may appear in a lot of reactions related to VOC oxidation.
Line 157. In general, two species should have shorter distances if they appear together in multiple reactions (e.g. NO and NO₂, HO and HO₂) or have similar products in the mechanism.

*Reviewer 1 on Line 285: Specify exactly which index denotes product and which denotes educt, i.e. "… include both educt i and product j species".*
**Response**. Sorry, we didn't make it clear. Either species i or j could be the reactant.
Line 164. Where $T_{(i, j)}$ is the number of reactions that include both species i and j (with one as reactant and the other as product, **either *i* or *j* could be the reactant**).

*Reviewer 2 on Line 465: Copy-pasted responses do not respect reviewers and editors time invested in this manuscript. I consider it a mauvais ton, especially as this happens not the first time in this submission.*
**Response**. Sorry, I did this because reviewers have similar questions. It is not a good manner.

*Reviewer 1 on Line 209: Do you "think" that there is more active iodine in the Block 3, or do you "believe" in that, or do you postulate that? Can you actually look into the model results to confirm that?*
**Response**. We indeed looked into the model results. Please see the details here.

Percentage of gridboxes in the global tropospheric+stratospheric domain that treat this species as fast.
**Block 3**: BrNO2(16.5%), IONO(9.5%), OIO(28.9%), ClOO(78.6%), OClO(20.6%), BrCl(25.5%), HOI(35.1%), Br2(28.2%), IONO2(24.7%), BrNO3(40.2%), I(51.1%), IO(50.7%), HOBr(48.6%), HOCl(34.4%), ClNO3(30.8%), BrO(51.3%), HCl(46%), HBr(26.9%), Cl(79%), Br(51.4%), ClO(49.1%)

**Block 4**: AERI(0.2%), ISALA(0.1%), ISALC(0.2%), I2O4(1.2%), I2O2(4.7%), I2O3(0.9%), IBr(0.7%), INO(0.9%), HI(1.2%), ICl(4.4%), Cl2O2(2.2%), ClNO2(2%), BrSALC(1.9%), BrSALA(2.2%), I2(2.3%), Cl2(2.5%)

---

## Editor Decision (ED2)

these new species following the diagram in Figure S8. After running the new version of the model for 12 months, our reduced algorithm shows consistent improvement in performance, reducing the chemical integration time by 53% and maintaining

295     error <0.5% in the boundary layer and 2-3% in the free troposphere and stratosphere (Figure 6c).

**4. Conclusions**

The high computational cost of chemical integration is a long-standing limitation in global atmospheric chemistry models. Typical chemical mechanisms include over 100 species coupled on short time scales. Previous research has proposed a variety of ways to speed up the chemical operator, all involving some loss of accuracy or generality. In this study, we have presented

300     a machine learning-guided adaptive method that can reduce the chemical integration time  while  and retaining full diagnostic capability.

In our algorithm, we first partition the mechanism species in into chemically logical blocks using a machine learning approach that analyzes production/loss rates and chemical linkages between species. We then assemble these blocks into an ensemble of submechanisms to encompass the range of chemical environments in the atmosphere. The model picks locally on the fly

305     which submechanism to use based on species' production and loss rates. The original mechanism can thus be greatly reduced in most environments while maintaining complexity where needed.  Updates to the original mechanism can be accommodated by assigning new species to the existing blocks without having to reconstruct the suite of submechanisms.

310     Our method has many advantages over previously proposed approaches to reduce chemical mechanism: (1) it is chemically logical; (2) it can save 50% computer time in chemical integration with errors lower than 2% for important species; (3) it is stable (no error growth over time) for 8-year simulations; (4) it retains full diagnostic information of concentration and rates; and (5) it is scale-independent. Our algorithm can significantly ease the computational bottleneck of chemical kinetics in global atmospheric chemistry models.

315     **Code availability**. The standard GEOS-Chem code is available through https://doi.org/10.5281/zenodo.1343547 (version 12.0.0) and https://doi.org/10.5281/zenodo.3950473 (version 12.9.1). The updates for the adaptive mechanism can be found at https://doi.org/10.7910/DVN/KASQOC.

**Data availability**. All datasets used in this study are publically accessible at https://doi.org/10.7910/DVN/KASQOC.

320     **Author contribution.** L. Shen and D. Jacob designed the experiments and L. Shen carried them out. L. Shen and D. Jacob prepared the manuscript with contributions from all co-authors.

---

## Author Response (AR3)

**Response to referee comments on "A machine learning-guided adaptive algorithm to reduce the computational cost of atmospheric chemistry in Earth System models: application to GEOS-Chem versions 12.0.0 and v12.9.1"**

We thank the editor for his careful reading of the manuscript and the valuable comments. This document is organized as follows: the editor's comments are in *italic*, our responses are in plain text, and all the revisions in the manuscript are shown in blue. **Boldface blue text** denotes text written in direct response to the Referee's comments. The line numbers in this document refer to the updated document with tracked changes.

**Summary of Comments on Microsoft Word - 3_reply_to_comments.docx**

**Page: 2**

**T** Number: 1     Author: gromov     Subject: Highlight     Date: 09/01/2022 15:23:40
Please change the vertical axis in panel (c) so that it presents the area between -1% and +1% at a greater magnification (you can use axis breaks). The most interesting results are in this area! (I confirm that after magnifying this figure in vector manipulation software)

Please see the comment regarding RRMSE below. Following it:
- Panel (a) shoud be removed,
- Analysis should be shown for at least the boundary layer, free troposphere and outside-of-troposphere domains.

**Response**. Thanks for these suggestions. The axis breaks didn't work well so we have a supplementary figure that specially presents the -1% and +1% area (Figure S6). We have removed panel (a) and show results for the boundary layer, free troposphere and stratosphere domain. We have updated Figure 6, added new Figure 7-8, S5-S8.We discuss these results in Line 310-355 (Section 3.2). Please check them for details.

[revised manuscript text omitted]

Exactly this statement has to be present in the manuscript. Because it explains how you came to this number!
* * *
**Response**. Thanks. We now say

Line 102. In each submechanism, if a reaction is slower than 10 molecules $cm^{-3}$ $s^{-1}$ for all gridboxes that select this submechanism, then the reaction is considered negligible and removed from the submechanism. The logic is that such a slow reaction will not contribute significantly to the total species production/loss rate threshold $\delta$ =500-1500 molecules $cm^{-3}$ $s^{-1}$. About 40-60% reactions can be removed using this strategy without incurring significant error. For example, reactions of short-lived volatile organic compounds (VOCs) are removed in stratospheric gridboxes, and daytime photochemical reactions are removed in nightime gridboxes. **Tests indicate that increasing the reaction rate threshold to 100 molecules $cm^{-3}$ $s^{-1}$ incurs significant error.**
* * *
**T** Number: 2     Author: gromov     Subject: Highlight     Date: 09/01/2022 20:51:35

Same as above, this statement has to be in the manuscript, augmented with the one mentioning that decreasing distances allows further optimisation without increasing the errors (what you formulate as "did not significantly change the results" highlighted below).
* * *
**Response**. Thanks. We now say

Line 144. Second, for each species *i*, we decrease its distance with the 5 species that have highest similarity with it by 50% and this scaling is applied only once for each species pair. The logic is that the number of species with similar chemical characteristics is usually around 5 and decreasing the distances among them by 50% can increase the probability of these species to be in the same chemical blocks after the optimization process. **We carried out a number of tests by perturbing the parameters used here and examine if the optimized chemical blocks are chemically logical, and results show that using 10 highest-similarity species instead of 5 or decreasing distances by 30%-70% instead of 50% did not significantly change the results.**

**T** Number: 3     Author: gromov     Subject: Highlight     Date: 09/01/2022 19:56:03

I believe you do may not correctly understand the meaning of the term "intuition" (look up the definition, e.g. at https://dictionary.cambridge.org/dictionary/english/intuition ). If your feelings determine which mechanism is correct, then your are submitting this study to a wrong journal -- GMD is dealing with natural sciences, foremost mathematics, physics and chemistry that are based on facts and logic.

Having said this, I admit that intuition is a subjective criterion and may precede an inference in physical sciences, however it may not be used as an argument in contrast to a strict mathematical/logical proof.

Please refrain from using this term. Also -- I repeat -- please give a clear definition (in the beginning, i.e. in the Introduction) to the "chemical coherence" term that you are using throughout the manuscript. "Coherence" in your case, I recon, means "having its parts related in an organized and reasonable way" (see definition at, e.g., https://dictionary.cambridge.org/dictionary/english/coherent ), so please explicate in which parts the mechanisms called "chemically coherent" are related in organised and reasonable way. In a way, this was somewhat satisfied in the sentence that you've edited (I refer to ll. 58-60 of the manuscript version 5 as compared to ver. 4). Please refrain from using "chemical coherence" in the abstract at all.

**Response**. Thanks for pointing this out. We have removed 'coherent' everywhere in the paper. In some places, we use 'chemically logical' and define it in the introduction.

Line 37. chemically logical (i.e., retaining connections between species involved in the same or similar reactions)

**T** Number: 4     Author: gromov     Subject: Highlight     Date: 04/01/2022 23:46:17

**Response**. Done.

**T** Number: 5     Author: gromov     Subject: Highlight     Date: 09/01/2022 20:37:47

I see improvements in Figs. 7 & 8, however I still do not see any relevance in what is shown S4 and S8. What if the most important species turn out to be those in >90% quantile part with RRMSE of 10% and higher? Which are those species that have highest RRMSE?

**Response**. Now we give a list of these species that have a mass bias or RRMSE in the >90% quantile part in table S1 and also discuss the results in main text.

Line 346. Table S1 lists the species with 10% highest RRMSE in each of the three atmospheric domains, dominated by secondary VOCs and iodine radicals in the troposphere, and VOC species in the stratosphere. None of these species play a central role in the chemistry for the corresponding atmospheric domains.

Line 354. Table S1 lists the species with 10% highest relative bias in atmospheric masses; all have minor importance in atmospheric chemistry.

Table S1. Top 10% Species with highest RRMSE and relative mass bias in the boundary layer, free troposphere and stratosphere.

| | RRMSE | Relative mass bias |
|---|---|---|
| Boundary layer | I2O4, I2O2, I2O3, IBr, INO, ICl, IONO, OIO, PRPN, HOI, BrSALC, BrSALA, ISNOHOO, IONO2, ISNOOB, I2, PRN1, MAN2, MACRN, MAOPO2, IO | I2O2, I2O3, INO, TRO2, N, XRO2, Cl2O2, IONO, OIO, PRPN, Br2, BrSALC, BrSALA, ISNOHOO, IONO2, ISNOOB, I2, Cl2, PRN1, MAN2, |

| | | MACRN, MAOPO2 |
|---|---|---|
| Free troposphere | ISN1OG, I2O4, I2O2, I2O3, INO, ICl, OIO, PRPN, BrSALA, MAOP, ISNOHOO, ISNOOB, I2, PRN1, MAN2, ISNOOA, MACRN, MAOPO2, I, IO, NMAO3, INO2 | AERI, ISN1OG, I2O4, I2O2, I2O3, INO, HI, IONO, OIO, HOI, BrSALC, BrSALA, ISNOHOO, ISNOOB, I2, Cl2, PRN1, MAN2, MACRN, I, IO, NMAO3 |
| Stratosphere | ISN1OA, ISN1OG, LVOC, PMNN, MRP, IPMN, MACRNO2, MONITS, GAOO, MVKN, MGLYOO, GLYX, MGLOO, MAN2, MACRN, HCOOH, KO2, MGLY, RIO2, INO2, MRO2 | ISN1OA, ISN1OG, DHDC, PRPN, DHPCARP, ISNOHOO, ISNOOB, INPN, I2, PRN1, PROPNN, MAN2, ISNOOA, MACRN, MAOPO2, OLND, OLNN, KO2, NMAO3, ISN1, RIO2, INO2 |
* * *
**T** Number: 6      Author: gromov      Subject: Highlight      Date: 09/01/2022 14:49:15

My comment was NOT about N2O, it was about OH. This answer is irrelevant.

**Response**. Yes, you are right that OH is so critical in atmospheric chemistry and a small bias in OH may have large effect on certain species. But the bias of OH is very small in our case (<0.2% in the in the troposphere and <0.01% in the stratosphere) (Figure S7) and there is no significant sign of increasing errors over our 8-year period (see Figure 7-8, S5-S7 for more details). We have removed all our old statements claiming that our algorithm can be applied in earth system models and make it clear that our error is stable in an 8-year period (we never over-state the performance of our method now). Now we say this in the text. Line 354. OH has a bias <0.2% in the troposphere and <0.01% in the stratosphere.

**Page: 4**

**T** Number: 1      Author: gromov      Subject: Highlight      Date: 09/01/2022 20:42:26

Relative differences for global atmospheric masses do not make any sense because species are not evenly distributed spatially and their concentrations change within orders of magnitude, also following the air density changes with height. What does an error of 10% for global mass tell me? Is it in the stratosphere, which has about 10% of all global mass, or at the surface? If the former, you have 200% error in the stratosphere?

See also the comments above regarding the calculation of RRMSE and presentation of errors for all species.

**Response**. Thanks for this good point. Now we show the error of boundary layer, free troposphere and stratosphere. Please see Figure 7-8, S5-S8 for more details. The related discussion can be found in Line 336-356.
* * *
**T** Number: 1     Author: gromov     Subject: Highlight     Date: 09/01/2022 20:30:13

I am still not fine with the averaging used in Eq (7) (instead of selection of maximum error across the gridboxes as in the cited literature). But, OK, then the only fairly acceptable form of threshold (a) for RRMSE you may define IS the relative one (i.e. 5th percentile that you use, I do not advise to put a lower limit at all; even if, then <=1 molecule cm^-3, as in classical approaches you refer to).

Therefore: please remove any results for "absolute" thresholds from entire manuscript - this will affect Figs. 6 to 8 and S4 to S8, if I did not miss anything. Choosing an absolute threshold value at this magnitude is not appropriate - the obtained RRMSE estimates are deceptive (e.g. at the value of 10^6 molec/cm3, only half of gridboxes with OH in your results will be accounted for on average). There are other species that have similar or even lower than OH concentrations in the atmosphere. Using fixed a may also create very large RRMSE values with large abundances, have you thought of this?

Furthermore, in the upper troposphere and stratosphere air density (and respective concentrations of trace gases with similar mixing ratios) is orders of magnitude lower. Therefore, please present the error analysis in Figure 8 for at least three atmospheric domains, viz. boundary layer (where most of "pollution chemistry" occurs), free troposphere (where, e.g., a major part of tropospheric of OH is) and outside the troposphere. This is adamant for such analysis. For Fig. 7 it may be enough to show boundary layer results. I hope that you are aware of the fact that the value of (a) should be chosen from the statistic on concentrations pertaining to the given domain, not all simulated gridboxes.

**Response**. Thanks for this good point. Now we have re-defined the RRMSE and we show the errors in the boundary layer, free troposphere and stratosphere.

Line 215. We use the Relative Root Mean Square Error (RRMSE) metric as given by Sandu et al. (1997) to characterize the error in our reduced mechanism:

$$RRMSE_i = \sqrt{\frac{1}{Q_i} \sum_{j=1}^{Q_i} \left( \frac{n_{i,j}^{reduced} - n_{i,j}^{full}}{n_{i,j}^{full}} \right)^2} \tag{7}$$

where $n_{i,j}^{reduced}$ and $n_{i,j}^{full}$ are the concentrations for species $i$ and gridbox $j$ in the reduced and full chemical mechanisms, and the sum is over the $Q_i$ ordered gridboxes that account for 99% of the global mass of species $i$, We calculate separate RRMSEs for the boundary layer (surface to 2 km), free troposphere (2 km to tropopause), and stratosphere with the same 99% threshold in each atmospheric domain. As a test, we also calculate the RRMSEs over the gridboxes that can account for 99% of the mass in each atmospheric domain (the 99% thresholds are different in different domains in this case).
* * *
**T** Number: 2     Author: gromov     Subject: Highlight     Date: 09/01/2022 15:05:52

A typo? What is chemical initiation???

**Response**. It is a typo. Fixed.
* * *
**T** Number: 3     Author: gromov     Subject: Highlight     Date: 04/01/2022 19:09:01

See the comment regarding the intuition above

**Response**. Thanks. We now use 'chemically logical' and define it in the introduction.

Line 37. chemically logical (i.e., retaining connections between species involved in the same or similar reactions)
* * *
🔲 Number: 4     Author: gromov     Subject: Highlight     Date: 09/01/2022 20:21:01
This is a very vague reply statement I cannot agree with unless you present solid arguments. Systematic biases are never welcome when estimates regarding any reactive (oxidant) species are made, e.g. not only CH4, a rather large number of trace gases are being oxidised by the HOx and NOx. An 8-yr simulation, I tend to agree here, will likely be sufficient to have reactive N-bearing reservoirs equilibrated.

**Response**. In our last replay statement, we say 'A 30-year simulation is not necessary for tropospheric oxidant/aerosol chemistry'. As seen from Figure S6, the mass bias for atmospheric oxidants is very small. For OH, the bias is <0.2% in the troposphere and <0.01% in the stratosphere. Even though this cannot rule out the possibility of increasing errors over a 30-year period, our 8-year simulation results imply that our algorithm works well in a timeframe that is suitable for many tropospheric atmospheric chemistry studies (e.g. air quality studies). In text, we make it clear that the error is stable in an 8-year simulation and we don't over-state our algorithm's performance. We hope these can ease your concern here.

Line 354. OH has a bias <0.2% in the troposphere and <0.01% in the stratosphere.
Line 449. It is stable (no error growth over time) for 8-year simulations.

**Page: 6**

🔲 Number: 1     Author: gromov     Subject: Highlight     Date: 09/01/2022 16:09:15
Sorry, you still haven't made if clear. "Either … could be" or "both … are" reactants? Reactants are those appearing on the LHS of the chemical equation. Products are on the RHS. Do indices refer to reactants only?

**Response**. Now we make it clear in text.
Line 128. Where $T_{i,j}$ is the number of reactions that include both species $i$ and $j$ (with $i$ as reactant and j as product, or $i$ as product and j as reactant)…

**Summary of Comments on Microsoft Word - 1_Manuscript_clean.docx**

**Page: 3**

🔲 Number: 1     Author: gromov     Subject: Highlight     Date: 09/01/2022 14:39:21
of GEOS-Chem?

**Response**. It is a typo. Fixed now.

**Page: 7**

T Number: 1    Author: gromov    Subject: Highlight    Date: 09/01/2022 15:02:19
Check the reference (w.r.t. that given in the References section), this metric was introduced in

A. Sandu, J.G. Verwer, M. Van Loon, G.R. Carmichael, F.A. Potra, D. Dabdub, J.H. Seinfeld. Benchmarking stiff ode solvers for atmospheric chemistry problems-I. implicit vs explicit. Atmospheric Environment, Volume 31, Issue 19, 1997, https://doi.org/10.1016/S1352-2310(97)00059-9.

**Response**. We have corrected it. Now we cite the Sandu's part I paper instead of part II.

**Page: 20**

T Number: 1    Author: gromov    Subject: Highlight    Date: 09/01/2022 20:02:17
Do you mean RRMSE?
Is this also calculated with a=10^6?

T Number: 2    Author: gromov    Subject: Highlight    Date: 09/01/2022 20:14:14

**Response**. We now use a different way to define RRMSE and have fixed the typo.

**Page: 21**

T Number: 1    Author: gromov    Subject: Highlight    Date: 09/01/2022 20:14:53
Eq. 7 ?

**Response**. It is a typo. Now fixed.

---

## Author Response (AR4)

**Response to referee comments on "A machine learning-guided adaptive algorithm to reduce the computational cost of atmospheric chemistry in Earth System models: application to GEOS-Chem versions 12.0.0 and v12.9.1"**

We thank the editor for his careful reading of the manuscript and the valuable comments. This document is organized as follows: the editor's comments are in *italic*, our responses are in plain text, and all the revisions in the manuscript are shown in blue. **Boldface blue text** denotes text written in direct response to the Referee's comments. The line numbers in this document refer to the updated document with tracked changes.

*There is still one last important point left to be addressed, however. You have introduced the use two RRMSE definitions now, with cutoff concentrations corresponding to 99% of the global- and domain-wise masses. Please remove the former one – this is the issue we have discussed earlier about using the same threshold concentration for the entire atmosphere, it is deceptive. Global threshold may not be used for separate atmospheric domains, full stop.*

**Response**. Thanks. We now remove all text and figures using the global threshold.

Line 181. We use the Relative Root Mean Square Error (RRMSE) metric as given by Sandu et al. (1997) to characterize the error in our reduced mechanism:

$$RRMSE_i = \sqrt{\frac{1}{Q_i} \sum_{j=1}^{Q_i} \left( \frac{n_{i,j}^{\text{reduced}} - n_{i,j}^{\text{full}}}{n_{i,j}^{\text{full}}} \right)^2} \qquad (7)$$

where $n_{i,j}^{\text{reduced}}$ and $n_{i,j}^{\text{full}}$ are the concentrations for species $i$ and gridbox $j$ in the reduced and full chemical mechanisms, and the sum is over the $Q_i$ ordered gridboxes **that account for 99% of the total mass of species $i$ in the boundary layer (surface to 2 km), free troposphere (2 km to tropopause), and stratosphere (the 99% thresholds are different in different atmospheric domains).**

[Figure]

**Figure 6**. **Performance and accuracy of the adaptive chemical mechanism**. We test the performance of the adaptive method by (A1) removing slow species ($P_i$ or $L_i > \delta$) and (A2) removing slow reactions (reaction rate < 10 molecules cm$^{-3}$ s$^{-1}$). Results are shown on the last day of 3-year simulations. The unit of $\delta$ is molecules cm$^{-3}$ s$^{-1}$. The performance is measured by the computing processor unit (CPU) time used by the chemical operator, and the accuracy is measured by the median relative root mean square error (RRMSE) for species concentrations using the full chemical mechanism in the boundary layer (0-2 km altitude), free troposphere (2 km to tropopause), and stratosphere. For (a) and (b), we use $\delta$ as 500 and 1500 molecules cm$^{-3}$ s$^{-1}$ in GEOS-Chem 12.0.0 that has 228 species and 724 reactions. For (c), we port the algorithm to GEOS-Chem 12.9.1 that has 262 species and 850 reactions. The number of blocks ($N$) is 13 and the number of chemical regimes is 21 (20 submechanisms (M=20) and one full mechanism).

*Fig. 6 shows RRMSE, however it is not clear, which one, please make sure that the domain-wise cutoff is being used. Your new Figure S5 (with domain-wise cutoffs) presents excellent results, so please use it the manuscript instead of Figure 7. The latter, together with the "global" threshold RRMSE use (I refer to lines 185-189 of the manuscript v.6) must be removed.*

**Response**. Thanks. We have removed the old Figure 7 and replaced it with Figure S5. All texts using the global threshold are removed now.

[Figure]

**Figure 7**. Accuracy of the adaptive reduced chemistry mechanism algorithm over an 8-year GEOS-Chem simulation using a threshold δ of 1500 molecules cm$^{-3}$ s$^{-1}$ to separate fast and slow species. We show the RRMSE in the (a) boundary layer, (b) free troposphere, and (c) stratosphere. Results are also shown for the median RRMSE across all species in the mechanism and more specifically the RRMSE for ozone, OH, NO$_2$, and sulfate.

---

## Author Response (AR5)

**Response to referee comments on "A machine learning-guided adaptive algorithm to reduce the computational cost of atmospheric chemistry in Earth System models: application to GEOS-Chem versions 12.0.0 and v12.9.1"**

*Thanks for submitting the final revision. Please revise the "Conclusions" section so that you do not reiterate the same message three times in three different paragraphs. See the suggestion for the corrections in the attached PDF.*

**Response**. We thank the editor for his careful reading and the constructive comments that have significantly improved this manuscript. We have removed the repeated sentences as suggested.